# gRNAde: Geometric Deep Learning for 3D RNA inverse design

**Chaitanya K. Joshi**[1], **Arian R. Jamasb**[2], **Ramon Viñas**[3], **Charles Harris**[1],
**Simon V. Mathis**[1], **Alex Morehead**[4], **Rishabh Anand**[5], **Pietro Liò**[1]

[1]University of Cambridge, UK, [2]Prescient Design, Genentech, Roche, [3]EPFL, Switzerland,
[3]University of Missouri, USA [5]National University of Singapore

Correspondence: chaitanya.joshi@cl.cam.ac.uk

## ABSTRACT

Computational RNA design tasks are often posed as inverse problems, where sequences are designed based on adopting a single desired secondary structure without considering 3D conformational diversity. We introduce **gRNAde**, a **g**eometric **RNA de**sign pipeline operating on 3D RNA backbones to design sequences that explicitly account for structure and dynamics. gRNAde uses a multi-state Graph Neural Network and autoregressive decoding to generates candidate RNA sequences conditioned on one or more 3D backbone structures where the identities of the bases are unknown. On a single-state fixed backbone re-design benchmark of 14 RNA structures from the PDB identified by Das et al. (2010), gRNAde obtains higher native sequence recovery rates (56% on average) compared to Rosetta (45% on average), taking under a second to produce designs compared to the reported hours for Rosetta. We further demonstrate the utility of gRNAde on a new benchmark of multi-state design for structurally flexible RNAs, as well as zero-shot ranking of mutational fitness landscapes in a retrospective analysis of a recent ribozyme. Experimental wet lab validation on 10 different structured RNA backbones finds that gRNAde has a success rate of 50% at designing pseudoknotted RNA structures, a significant advance over 35% for Rosetta. Open source code and tutorials are available at: github.com/chaitjo/geometric-rna-design

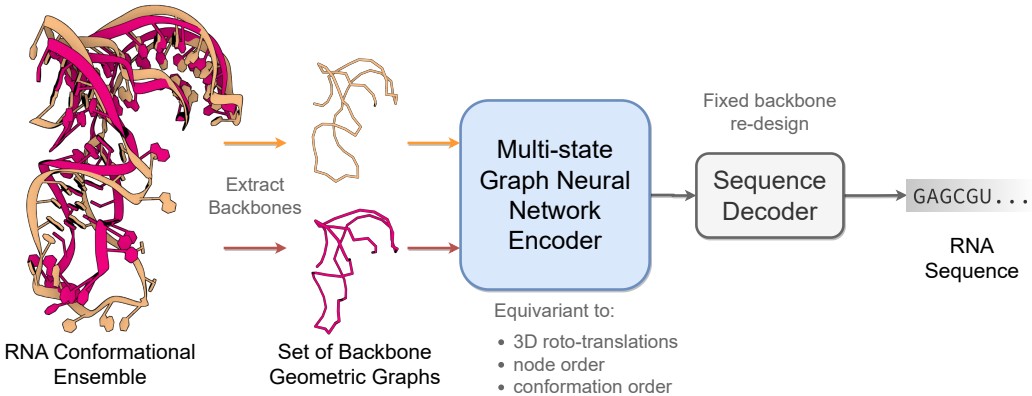

Figure 1: **The gRNAde pipeline for 3D RNA inverse design.** gRNAde is a generative model for RNA sequence design conditioned on backbone 3D structure(s). gRNAde processes one or more RNA backbone graphs (a conformational ensemble) via a multi-state GNN encoder which is equivariant to 3D roto-translation of coordinates as well as conformational state order, followed by conformational state order-invariant pooling and autoregressive sequence decoding.

## 1 INTRODUCTION

**Why RNA design?** Historical efforts in computational drug discovery have focussed on designing small molecule or protein-based medicines that either treat symptoms or counter the end stages of disease processes. In recent years, there is a growing interest in designing new RNA-based therapeutics that intervene earlier in disease processes to cut off disease-causing information flow in the cell (Damase et al., 2021; Zhu et al., 2022). Notable examples of RNA molecules at the forefront of biotechnology today include mRNA vaccines (Metkar et al., 2024) and CRISPR-based genomic medicine (Doudna & Charpentier, 2014). Of particular interest for structure-based design are ribozymes and riboswitches in the untranslated regions of mRNAs (Mandal & Breaker, 2004; Leppek et al., 2018). In addition to coding for proteins (such as the spike protein in the Covid vaccine), naturally occurring mRNAs contain riboswitches that are responsible for cell-state dependent protein expression of the mRNA. Riboswitches act by 'switching' their 3D structure from an unbound conformation to a bound one in the presence of specific metabolites or small molecules. Rational design of riboswitches will enable translation to be dependent on the presence or absence of partner molecules, essentially acting as 'on-off' switches for highly targeted mRNA therapies in the future (Felletti et al., 2016; Mustafina et al., 2019; Mohsen et al., 2023).

**Challenges of RNA modelling.** Despite the promises of RNA therapeutics, proteins have been the primary focus in the 3D biomolecular modelling community. Availability of a large number of protein structures from the PDB combined with advances in deep learning for structured data (Bronstein et al., 2021; Duval et al., 2023) have revolutionized protein 3D structure prediction (Jumper et al., 2021) and rational design (Dauparas et al., 2022; Watson et al., 2023). Applications of deep learning for RNA design are underexplored compared to proteins due to paucity of 3D structural data (Schneider et al., 2023). Most tools for RNA design primarily focus on secondary structure without considering 3D geometry (Churkin et al., 2018) or use non-learnt algorithms for aligning 3D RNA fragments (Han et al., 2017; Yesselman et al., 2019), which can be restrictive due to the hand-crafted nature of the heuristics used. In addition to limited 3D data, the key technical challenge is that RNA is generally more dynamic than proteins. The same RNA can adopt multiple distinct conformational states to create and regulate complex biological functions (Ganser et al., 2019; Hoetzel & Suess, 2022; Ken et al., 2023). Computational RNA design pipelines must account for both the 3D geometric structure and conformational flexibility of RNA to engineer new biological functions.

**Our contributions.** This paper introduces **gRNAde**, a **g**eometric deep learning-based pipeline for **RNA** inverse **de**sign. As illustrated in Figure 1, gRNAde generates candidate RNA sequences conditioned on one or more 3D backbone structures, enabling both single- and multi-state fixed-backbone sequence design. We demonstrate the utility of gRNAde for the following design scenarios:

- *Improved performance and speed over Rosetta.* We compare gRNAde to Rosetta (Leman et al., 2020), the state-of-the-art physics-based tool for 3D RNA inverse design, for single-state fixed backbone design of 14 RNA structures of interest from the PDB identified by Das et al. (2010). We obtain higher native sequence recovery rates with gRNAde (56% on average) compared to Rosetta (45% on average). Additionally, gRNAde is significantly faster than Rosetta for inference; e.g. sampling 100+ designs in 1 second for an RNA of 60 nucleotides on an A100 GPU (<10 seconds on CPU), compared to the reported hours for Rosetta on CPU.

- *Multi-state RNA design*, which was previously not possible with Rosetta. gRNAde with multi-state GNNs improves sequence recovery by 5% over an equivalent single-state model on a benchmark of structurally flexible RNAs, especially for surface nucleotides which undergo positional or secondary structural changes. gRNAde's GNN is the first geometric deep learning architecture for multi-state biomolecule representation learning.

- *Zero-shot learning of RNA fitness landscape.* In a retrospective analysis of mutational fitness landscape data for an RNA polymerase ribozyme (McRae et al., 2024), we show how gRNAde's perplexity, the likelihood of a sequence folding into a backbone structure, can be used to rank mutants based on fitness in a zero-shot/unsupervised manner and outperforms random mutagenesis for improving fitness over the wild type in low throughput scenarios.

- *Wet lab validated.* As part of Eterna's OpenKnot Round 6, 200 gRNAde-designed RNAs for diverse backbones were independently validated in a wet lab via SHAPE chemical mapping experiments. gRNAde demonstrated an overall success rate of 50% at designing RNAs with desired pseudoknotted structures, which is a significant improvement over Rosetta with 35%.

## 2 THE GRNADE PIPELINE

Figure 1 illustrates the RNA inverse folding problem: the task of designing new RNA sequences conditioned on a structural backbone. Given the 3D coordinates of a backbone structure, machine learning models must generate sequences that are likely to fold into that shape. The underlying assumption behind inverse folding (and rational biomolecule design) is that structure determines function (Huang et al., 2016).

Following best practices in protein design, gRNAde uses a structure-conditioned, autoregressive language model with geometric GNN encoder and decoder (Jing et al., 2020; Dauparas et al., 2022). Our main architectural contribution is a multi-state GNN for modelling sets of 3D backbones (described in Section 2.2) as well as an efficient PyG implementation (Appendix Figure 12 and the pseudocode). To the best of our knowledge, gRNAde is the first explicitly multi-state inverse folding pipeline, allowing users to design sequences for backbone conformational ensembles (a set of 3D backbone structures) as opposed to a single structure.

### 2.1 RNA CONFORMATIONAL ENSEMBLES AS GEOMETRIC MULTI-GRAPHS

**Featurization.** The input to gRNAde is an RNA to be re-designed. For instance, this could be a set of PDB files with 3D backbone structures for the given RNA (a conformational ensemble) and the corresponding sequence of $n$ nucleotides. As shown in Appendix Figure 9, gRNAde builds a geometric graph representation for each input structure:

1. We start with a 3-bead coarse-grained representation of the RNA backbone, retaining the coordinates for P, C4', N1 (pyrimidine) or N9 (purine) for each nucleotide (Dawson et al., 2016). This 'pseudotorsional' representation describes RNA backbones completely in most cases while reducing the size of the torsional space to prevent overfitting (Wadley et al., 2007).

2. Each nucleotide $i$ is assigned a node in the geometric graph with the 3D coordinate $\vec{x}_i \in \mathbb{R}^3$ corresponding to the centroid of the 3 bead atoms. Random Gaussian noise with standard deviation 0.1Å is added to coordinates during training to prevent overfitting on crystallisation artifacts, following Dauparas et al. (2022). Each node is connected by edges to its 32 nearest neighbours as measured by the pairwise distance in 3D space, $\|\vec{x}_i - \vec{x}_j\|_2$.

3. Nodes are initialized with geometric features analogous to the featurization used in protein inverse folding (Ingraham et al., 2019; Jing et al., 2020): (a) forward and reverse unit vectors along the backbone from the 5' end to the 3' end, ($\vec{x}_{i+1} - \vec{x}_i$ and $\vec{x}_i - \vec{x}_{i-1}$); and (b) unit vectors, distances, angles, and torsions from each C4' to the corresponding P and N1/N9.

4. Edge features from node $j$ to $i$ are initialized as: (a) the unit vector from the source to destination node, $\vec{x}_j - \vec{x}_i$; (b) the distance in 3D space, $\|\vec{x}_j - \vec{x}_i\|_2$, encoded by 32 radial basis functions; and (c) the distance along the backbone, $j - i$, encoded by 32 sinusoidal positional encodings.

**Multi-graph representation.** As described in the previous section, given a set of $k$ (conformational state) structures in the input conformational ensemble, each RNA backbone is featurized as a separate geometric graph $\mathcal{G}^{(k)} = (\boldsymbol{A}^{(k)}, \boldsymbol{S}^{(k)}, \vec{\boldsymbol{V}}^{(k)})$ with the scalar features $\boldsymbol{S}^{(k)} \in \mathbb{R}^{n \times f}$, vector features $\vec{\boldsymbol{V}}^{(k)} \in \mathbb{R}^{n \times f' \times 3}$, and $\boldsymbol{A}^{(k)}$, an $n \times n$ adjacency matrix. For clear presentation and without loss of generality, we omit edge features and use $f$, $f'$ to denote scalar/vector feature channels.

The input to gRNAde is thus a set of geometric graphs $\{\mathcal{G}^{(1)}, \ldots, \mathcal{G}^{(k)}\}$ which is merged into a 'multi-graph' representation of the conformations, $\mathcal{M} = (\boldsymbol{A}, \boldsymbol{S}, \vec{\boldsymbol{V}})$, by stacking the set of scalar features $\{\boldsymbol{S}^{(1)}, \ldots, \boldsymbol{S}^{(k)}\}$ into one tensor $\boldsymbol{S} \in \mathbb{R}^{n \times k \times f}$ along a new axis for the set size $k$. Similarly, the set of vector features $\{\vec{\boldsymbol{V}}^{(1)}, \ldots, \vec{\boldsymbol{V}}^{(k)}\}$ is stacked into one tensor $\vec{\boldsymbol{V}} \in \mathbb{R}^{n \times k \times f' \times 3}$. Lastly, the set of adjacency matrices $\{\boldsymbol{A}^{(1)}, \ldots, \boldsymbol{A}^{(k)}\}$ are merged via union $\cup$ into a joint adjacency matrix $\boldsymbol{A}$.

### 2.2 MULTI-STATE GNN FOR ENCODING CONFORMATIONAL ENSEMBLES

The gRNAde model, illustrated in Appendix Figure 10, processes one or more RNA backbone graphs via a multi-state GNN encoder which is equivariant to 3D roto-translation of coordinates as well as to the ordering of conformational states, followed by conformational state order-invariant pooling and sequence decoding. We describe each component in the following sections.

**Multi-state GNN encoder.** When representing conformational ensembles as a multi-graph, each node feature tensor contains three axes: (#nodes, #conformations, feature channels). We perform message passing on the multi-graph adjacency to *independently* process each conformational state, while maintaining permutation equivariance of the updated feature tensors along both the first (#nodes) and second (#conformations) axes. This works by operating on only the feature channels axis and generalising the PyTorch Geometric (Fey & Lenssen, 2019) message passing class to account for the extra conformations axis; see Appendix Figure 12 and the pseudocode for details.

We use multiple rotation-equivariant GVP-GNN (Jing et al., 2020) layers to update scalar features $\boldsymbol{s}_i \in \mathbb{R}^{k \times f}$ and vector features $\vec{\boldsymbol{v}}_i \in \mathbb{R}^{k \times f' \times 3}$ for each node $i$:

$$\boldsymbol{m}_i, \vec{\boldsymbol{m}}_i := \sum_{j \in \mathcal{N}_i} \mathrm{MSG}\big( \left( \boldsymbol{s}_i, \vec{\boldsymbol{v}}_i \right), \left( \boldsymbol{s}_j, \vec{\boldsymbol{v}}_j \right), e_{ij} \big), \tag{1}$$

$$\boldsymbol{s}_i', \vec{\boldsymbol{v}}_i' := \mathrm{UPD}\big( \left( \boldsymbol{s}_i, \vec{\boldsymbol{v}}_i \right) \, , \, \left( \boldsymbol{m}_i, \vec{\boldsymbol{m}}_i \right) \big), \tag{2}$$

where $\mathrm{MSG}, \mathrm{UPD}$ are Geometric Vector Perceptrons, a generalization of MLPs to take tuples of scalar and vector features as input and apply $O(3)$-equivariant non-linear updates. The overall GNN encoder is $SO(3)$-equivariant due to the use of reflection-sensitive input features (dihedral angles) combined with $O(3)$-equivariant GVP-GNN layers.

Our multi-state GNN encoder is easy to implement in any message passing framework and can be used as a *plug-and-play* extension for any geometric GNN pipeline to incorporate the multi-state inductive bias. It serves as an elegant alternative to batching all the conformations, which we found required major alterations to message passing and pooling depending on downstream tasks.

**Conformation order-invariant pooling.** The final encoder representations in gRNAde account for multi-state information while being invariant to the permutation of the conformational ensemble. To achieve this, we perform a Deep Set pooling (Zaheer et al., 2017) over the conformations axis after the final encoder layer to reduce $\boldsymbol{S} \in \mathbb{R}^{n \times k \times f}$ and $\vec{\boldsymbol{V}} \in \mathbb{R}^{n \times k \times f' \times 3}$ to $\boldsymbol{S}' \in \mathbb{R}^{n \times f}$ and $\vec{\boldsymbol{V}}' \in \mathbb{R}^{n \times f' \times 3}$:

$$\boldsymbol{S}', \vec{\boldsymbol{V}}' := \frac{1}{k} \sum_{i=1}^{k} \left( \boldsymbol{S}[:, \, i], \vec{\boldsymbol{V}}[:, \, i] \right). \tag{3}$$

A simple sum or average pooling does not introduce any new learnable parameters to the pipeline and is flexible to handle a variable number of conformations, enabling both single-state and multi-state design with the same model. In Appendix B, we also explore more expressive geometric set pooling functions (Maron et al., 2020).

**Sequence decoding and loss function.** We feed the final encoder representations after pooling, $\boldsymbol{S}', \vec{\boldsymbol{V}}'$, to autoregressive GVP-GNN decoder layers to predict the probability of the four possible base identities (A, G, C, U) for each node/nucleotide. Decoding proceeds according to the RNA sequence order from the 5' end to 3' end. gRNAde is trained in a self-supervised manner by minimising a cross-entropy loss (with label smoothing value of 0.05) between the predicted probability distribution and the ground truth identity for each base. During training, we use autoregressive teacher forcing (Williams & Zipser, 1989) where the ground truth base identity is fed as input to the decoder at each step, encouraging the model to stay close to the ground-truth sequence.

**Sampling.** When using gRNAde for inference and designing new sequences, we iteratively sample the base identity for a given nucleotide from the predicted conditional probability distribution, given the partially designed sequence up until that nucleotide/decoding step. We can modulate the smoothness or sharpness of the probability distribution by using a temperature parameter. gRNAde can also use unordered decoding (Dauparas et al., 2022) with minimal impact on performance, as well as masking or logit biasing during sampling, depending on the design scenario at hand.

### 2.3 Evaluation metrics for designed sequences

In principle, inverse folding models can be sampled from to obtain a large number of designed sequences for a given backbone structure. Thus, in-silico metrics to determine which sequences are useful and which ones to prioritise in wet lab experiments are a critical part of the overall pipeline. We currently use the following metrics to evaluate gRNAde's designs, visualised in Appendix Figure 11:

- **Native sequence recovery**, which is the average percentage of native (ground truth) nucleotides correctly recovered in the sampled sequences. Recovery is the most widely used metric for biomolecule inverse design (Dauparas et al., 2022) but can be misleading in the case of RNAs where alternative nucleotide base pairings can form the same structural patterns.

- **Secondary structure self-consistency score**, where we 'forward fold' the sampled sequences using a secondary structure prediction tool (we used EternaFold (Wayment-Steele et al., 2022)) and measure the average Matthew's Correlation Coefficient (MCC) to the groundtruth secondary structure, represented as a binary adjacency matrix. MCC values range between -1 and +1, where +1 represents a perfect match, 0 an average random prediction and -1 an inverse prediction. This measures how well the designs recover base pairing patterns.

- **Tertiary structure self-consistency scores**, where we 'forward fold' the sampled sequences using a 3D structure prediction tool (we used RhoFold (Shen et al., 2022)) and compute the average RMSD, TM-score and GDT_TS to the groundtruth C4' coordinates to measure how well the designs recover global structural similarity and 3D conformations.

- **Perplexity**, which can be thought of as the average number of bases that the model is selecting from for each nucleotide. Formally, perplexity is the average exponential of the negative log-likelihood of the sampled sequences. A 'perfect' model which regurgitates the groundtruth[1] would have perplexity of 1, while a perplexity of 4 means that the model is making random predictions (the model outputs a uniform probability over 4 possible bases). Perplexity does not require a ground truth structure to calculate, and can also be used for ranking sequences as it is the model's estimate of the compatibility of a sequence with the input backbone structure.

**Significance and limitations.** Self-consistency metrics, termed 'designability' (eg. scRMSD$\leq$2Å), as well as perplexity have been found to correlate with experimental success in protein design (Watson et al., 2023). While precise designability thresholds are yet to be established for RNA, pairs of structures with TM-score$\geq$0.45 or GDT_TS$\geq$0.5 are known to correspond to roughly the same fold (Zhang et al., 2022). Another major limitation for in-silico evaluation of 3D RNA design compared to proteins is the relatively worse state of structure prediction tools (Schneider et al., 2023).

## 3 EXPERIMENTAL SETUP

**3D RNA structure dataset.** We create a machine learning-ready dataset for RNA inverse design using RNASolo (Adamczyk et al., 2022), a novel repository of RNA 3D structures extracted from solo RNAs, protein-RNA complexes, and DNA-RNA hybrids in the PDB. We used all currently known RNA structures at resolution $\leq$4.0Å resulting in 4,223 unique RNA sequences for which a total of 12,011 structures are available (RNASolo date cutoff: 31 October 2023). As inverse folding is a per-node/per-nucleotide level task, our training data contains over 2.8 Million unique nucleotides. Further dataset statistics are available in Appendix Figure 13, illustrating the diversity of our dataset in terms of sequence length, number of structures per sequence, as well as structural variations among conformations per sequence.

**Structural clustering.** In order to ensure that we evaluate gRNAde's generalization ability to novel RNAs, we cluster the 4,223 unique RNAs into groups based on structural similarity. We use US-align (Zhang et al., 2022) with a similarity threshold of TM-score >0.45 for clustering, and ensure that we train, validate and test gRNAde on structurally dissimilar clusters (see next paragraph). We also provide utilities for clustering based on sequence homology using CD-HIT (Fu et al., 2012), which leads to splits containing biologically dissimilar clusters of RNAs.

**Splits to evaluate generalization.** After clustering, we split the RNAs into training ($\sim$4000 samples), validation and test sets (100 samples each) to evaluate two different design scenarios:

1. **Single-state split.** This split is used to fairly evaluate gRNAde for single-state design on a set of RNA structures of interest from the PDB identified by Das et al. (2010), which mainly includes riboswitches, aptamers, and ribozymes. We identify the structural clusters belonging to the RNAs identified in Das et al. (2010) and add all the RNAs in these clusters to the test set (100 samples). The remaining clusters are randomly added to the training and validation splits.

---

[1]Note that such a model would be practically useless for real design tasks.

2. **Multi-state split.** This split is used to test gRNAde's ability to design RNA with multiple distinct conformational states. We order the structural clusters based on median intra-sequence RMSD among available structures within the cluster[2]. The top 100 samples from clusters with the highest median intra-sequence RMSD are added to the test set. The next 100 samples are added to the validation set and all remaining samples are used for training.

Validation and test samples come from clusters with at most 5 unique sequences, in order to ensure diversity. Any samples that were not assigned clusters are directly appended to the training set. We also directly add very large RNAs (> 1000 nts) to the training set, as it is unlikely that we want to design very large RNAs. We exclude very short RNA strands (< 10 nts).

**Evaluation metrics.** For a given data split, we evaluate models on the held-out test set by designing 16 sequences (sampled at temperature 0.1) for each test data point and computing averages for each of the metrics described in Section 2.3: native sequence recovery, structural self-consistency scores and perplexity. We employ early stopping by reporting test set performance for the model checkpoint for the epoch with the best validation set recovery. Standard deviations are reported across 3 consistent random seeds for all models.

**Hyperparameters.** All models use 4 encoder and 4 decoder GVP-GNN layers, with 128 scalar/16 vector node features, 64 scalar/4 vector edge features, and drop out probability 0.5, resulting in 2,147,944 trainable parameters. All models are trained for a maximum of 50 epochs using the Adam optimiser with an initial learning rate of 0.0001, which is reduced by a factor 0.9 when validation performance plateaus with patience of 5 epochs. Ablation studies of key modelling decisions are available in Appendix Table 1.

## 4 RESULTS

### 4.1 SINGLE-STATE RNA DESIGN BENCHMARK

We set out to compare gRNAde to Rosetta, a state-of-the-art physics-based toolkit for biomolecular modelling and design (Leman et al., 2020). We reproduced the benchmark setup from Das et al. (2010) for Rosetta's fixed backbone RNA sequence design workflow on 14 RNA structures of interest from the PDB, which mainly includes riboswitches, aptamers, and ribozymes (full listing in Table 2). We trained gRNAde on the single-state split detailed in Section 3, explicitly excluding the 14 RNAs as well as any structurally similar RNAs in order to ensure that we fairly evaluate gRNAde's generalization abilities vs. Rosetta.

**gRNAde improves sequence recovery over Rosetta.** In Figure 2, we compare gRNAde's native sequence recovery for single-state design with numbers taken from Das et al. (2010) for Rosetta, FARNA (a predecessor of Rosetta), ViennaRNA (the most popular 2D inverse folding method), and RDesign (Tan et al., 2023). RDesign is a concurrent deep learning-based 3D inverse folding model which uses invariant GNN layers and non-autoregressive decoding without sampling; see Appendix F for details. gRNAde has higher recovery of 56% on average compared to 45% for Rosetta, 32% for FARNA, 27% for ViennaRNA, and 43% for RDesign.

**gRNAde is significantly faster than Rosetta.** In addition to superior sequence recovery, gRNAde is significantly faster than Rosetta for high-throughout design pipelines. Training gRNAde from scratch takes roughly 2–6 hours on a single A100 GPU, depending on the exact hyperparameters. Once trained, gRNAde can design hundreds of sequences for backbones with hundreds of nucleotides in ∼10 seconds on CPU and ∼1 second with GPU acceleration. On the other hand, Rosetta takes order of hours to produce a single design due to performing expensive Monte Carlo optimisation until convergence on CPU[3]. Deep learning methods like gRNAde are arguably easier to use since no expert customization is required and setup is easier compared to Rosetta (the latest builds do not include RNA recipes), making RNA design more broadly accessible.

---

[2]For each RNA sequence, we compute the pairwise C4' RMSD among all available structures. We then compute the median RMSD across all sequences within each structural cluster.

[3]We note that Rosetta documentation states that "runs on RNA backbones longer than ∼ten nucleotides take many minutes or hours". We have not run Rosetta ourselves as recent builds do not include RNA recipes. It is hard to fairly compare inference time as Rosetta recipes are not GPU-accelerated, while gRNAde is.

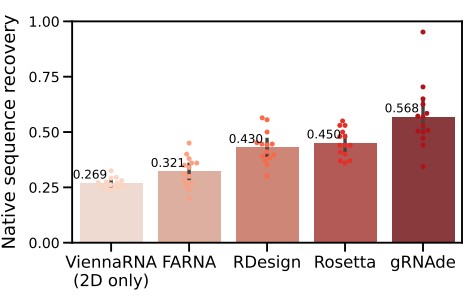

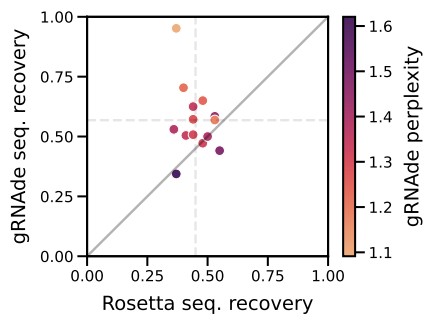

(a) gRNAde outperforms Rosetta.

(b) gRNAde's perplexity correlates with recovery.

Figure 2: **gRNAde compared to Rosetta for single-state design.** (a) We benchmark native sequence recovery of gRNAde, RDesign, Rosetta, FARNA and ViennaRNA on 14 RNA structures of interest identified by Das et al. (2010). gRNAde obtains higher native sequence recovery rates (56% on average) compared to Rosetta (45%) and all other methods. (b) Sequence recovery per sample for Rosetta and gRNAde, shaded by gRNAde's perplexity for each sample. gRNAde's perplexity is correlated with native sequence recovery for designed sequences (Pearson correlation: -0.76, Spearman correlation: -0.67). Full results on single-state test set are available in Appendix B and per-RNA results in Appendix Table 2.

**gRNAde's perplexity correlates with sequence recovery.** In Figure 2b, we plot native sequence recovery per sample for Rosetta vs. gRNAde, shaded by gRNAde's average perplexity for each sample. Perplexity is an indicator of the model's confidence in its own prediction (lower perplexity implies higher confidence) and appears to be correlated with native sequence recovery. In the subsequent Section 4.3, we further demonstrate the utility of gRNAde's perplexity for zero-shot ranking of RNA fitness landscapes.

## 4.2 MULTI-STATE RNA DESIGN BENCHMARK

Structured RNAs often adopt multiple distinct conformational states to perform biological functions (Ken et al., 2023). For instance, riboswitches adopt at least two distinct functional conformations: a ligand bound (holo) and unbound (apo) state, which helps them regulate and control gene expression (Stagno et al., 2017). If we were to attempt single-state inverse design for such RNAs, each backbone structure may lead to a different set of sampled sequences. It is not obvious how to select the input backbone as well as designed sequence when using single-state models for multi-state design. gRNAde's multi-state GNN, descibed in Section 2.2, directly 'bakes in' the multi-state nature of RNA into the architecture and designs sequences explicitly conditioned on multiple states.

In order to evaluate gRNAde's multi-state design capabilities, we trained equivalent single-state and multi-state gRNAde models on the multi-state split detailed in Section 3, where the validation and test sets contain progressively more structurally flexible RNAs as measured by median RMSD among multiple available states for an RNA.

**Multi-state gRNAde consistently boosts sequence recovery.** In Figure 3a, we compared a single-state variant of gRNAde with otherwise equivalent multi-state models (with up to 5 states) in terms of native sequence recovery. Multi-state variants show a consistent 3-5% improvement, with the best performance obtained using 3 states. This trend holds to a lesser extent on the single-state benchmark where the multi-state model is being used with only one state as input. This suggests that seeing multiple states during training can be useful for teaching gRNAde about RNA conformational flexibility and improve performance even for single-state design tasks. As a caveat, it is worth noting that multi-state models consume more GPU memory than an equivalent single-state model during mini-batch training (approximate peak GPU usage for max. number of states = 1: 12GB, 3: 28GB, 5: 50GB on a single A100 with at most 3000 total nodes in a mini-batch).

**Improved recovery in structurally flexible regions.** In Figure 3b, we evaluated gRNAde's multi-state sequence recovery at a fine-grained, per-nucleotide level to understand the source of performance gains. Multi-state GNNs improve sequence recovery over the single-state variant on structurally

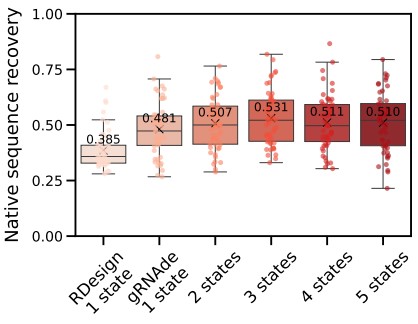 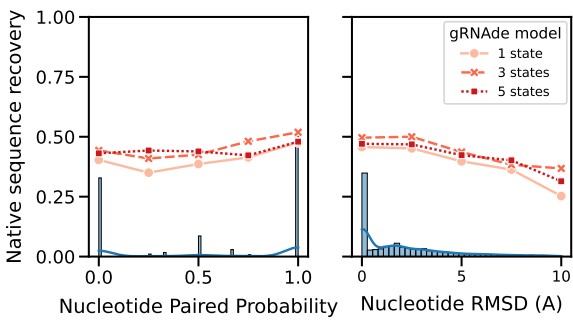

(a) Per-sample sequence recovery            (b) Per-nucleotide recovery vs. structural flexibility

Figure 3: **Multi-state design benchmark.** (a) Multi-state gRNAde shows a consistent 3-5% improvement over the single-state variant in terms of sequence recovery on the multi-state test set of 100 RNAs, with the best performance obtained using 3 states. (b) When plotting sequence recovery per-nucleotide, multi-state gRNAde improves over a single-state model for structurally flexible regions of RNAs, as characterised by nucleotides that tend to undergo changes in base pairing (left) and nucleotides with higher average RMSD across multiple states (right). Marginal histograms in blue show the distribution of values. We plot performance for one consistent random seed across all models; collated results and ablations are available in Appendix B.

flexible nucleotides, as characterised by undergoing changes in base pairing/secondary structure and higher average RMSD between 3D coordinates across states.

### 4.3 ZERO-SHOT RANKING OF RNA FITNESS LANDSCAPE

Lastly, we explored the use of gRNAde as a zero-shot ranker of mutants in RNA engineering campaigns. Given the backbone structure of a wild type RNA of interest as well as a candidate set of mutant sequences, we can compute gRNAde's perplexity of whether a given sequence folds into the backbone structure. Perplexity is inversely related to the likelihood of a sequence conditioned on a structure, as described in Section 2.3. We can then rank sequences based on how 'compatible' they are with the backbone structure in order to select a subset to be experimentally validated in wet labs.

**Retrospective analysis on ribozyme fitness landscape.** A recent study by McRae et al. (2024) determined a cryo-EM structure of a dimeric RNA polymerase ribozyme at 5Å resolution[4], along with fitness landscapes of ∼75K mutants for the catalytic subunit 5TU and ∼48K mutants for the scaffolding subunit t1. We design a retrospective study using this data of (sequence, fitness value) pairs where we simulate an RNA engineering campaign with the aim of improving catalytic subunit fitness over the wild type 5TU sequence.

We consider various design budgets ranging from hundreds to thousands of sequences selected for experimental validation, and compare 4 unsupervised approaches for ranking/selecting variants: (1) random choice from all ∼75,000 sequences; (2) random choice from all 449 single mutant sequences; (3) random choice from all single and double mutant sequences (as sequences with higher mutation order tend to be less fit); and (4) negative gRNAde perplexity (lower perplexity is better). For each design budget and ranking approach, we compute the expected maximum change in fitness over the wild type that could be achieved by screening as many variants as allowed in the given design budget. We run 10,000 simulations to compute confidence intervals for the 3 random baselines.

**gRNAde outperforms random baselines in low design budget scenarios.** Figure 4 illustrates the results of our retrospective study. At low design budgets of up to hundreds of sequences, which are relevant in the case of a low throughput fitness screening assay, gRNAde outperforms all random baselines in terms of the maximum change in fitness over the wild type. The top 10 mutants as ranked by gRNAde contain a sequence with 4-fold improved fitness, while the top 200 leads to a 5-fold improvement. Note that gRNAde is used zero-shot here, i.e. it was not fine-tuned on any assay data.

---

[4]This RNA was not present in gRNAde's training data, which contains structures at ≤4.0Å resolution.

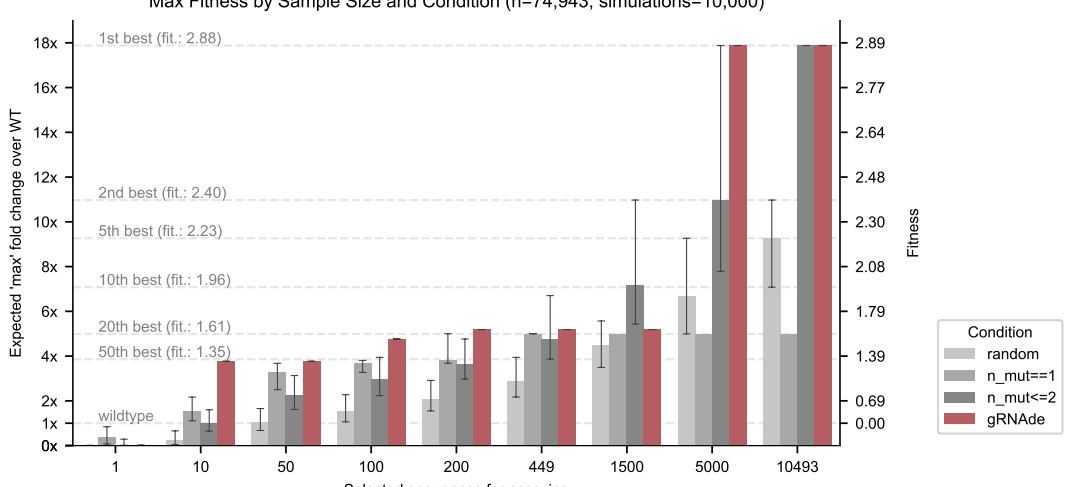

Figure 4: **Retrospective study of gRNAde for ranking ribozyme mutant fitness.** Using the backbone structure and mutational fitness landscape data from an RNA polymerase ribozyme (McRae et al., 2024), we retrospectively analyse how well we can rank variants at multiple design budgets using random selection vs. gRNAde's perplexity for mutant sequences conditioned on the backbone structure (catalytic subunit 5TU). Note that gRNAde is used zero-shot here, i.e. it was not fine-tuned on any assay data. For stochastic strategies, bars indicate median values, and error bars indicate the interquartile range estimated from 10,000 simulations per strategy and design budget. At low throughput design budgets of up to ∼500 sequences, selecting mutants using gRNAde outperforms random baselines in terms of the expected maximum improvement in fitness over the wild type. In particular, gRNAde performs better than single site saturation mutagenesis, even when all single mutants are explored (total of 449 single mutants, 10,493 double mutants for the catalytic subunit 5TU in McRae et al. (2024)). See Appendix Figure 7 for results on scaffolding subunit t1.

Overall, it is promising that gRNAde's perplexity correlates with experimental fitness measurements out-of-the-box (zero-shot) and can be a useful ranker of mutant fitness in our retrospective study. In realistic design scenarios, improvements could likely be obtained by fine-tuning gRNAde on a low amount of experimental fitness data. For example, latent features from gRNAde may be finetuned or used as input to a prediction head with supervised learning on fitness landscape data.

## 5 INDEPENDENT WET LAB VALIDATION OF GRNADE DESIGNS

Finally, we present the results of independent wet lab validation of gRNAde via Eterna, an online platform for computational RNA design. Eterna regularly releases new RNA design tasks to a global community of researchers and citizen-scientists who design sequences using expert intuition or computational tools (such as gRNAde and Rosetta). The designs are then experimentally validated at Stanford University via high-throughput SHAPE assays which measures the reactivity of an RNA sequence to a chemical modifier, as described in He et al. (2024).

**Eterna OpenKnot Round 6.** As part of OpenKnot Round 6, we submitted gRNAde designs for 10 target RNA backbones: SARS-CoV-2 frame shift element, Tetrahydrofolate riboswitch, GMP-II riboswitch, SAM riboswitch, HCV internal ribosome entry site, synthetic kissing loop structure, donggang dumbell, telomerase ribozyme, HDV ribozyme, and CPEB3 ribozyme. We submitted a total of 20 designs for each backbone via two approaches: (1) 10 partial designs, where parts of the wildtype sequence are kept fixed; and (2) 10 full designs, where the entire sequence is designed. In addition to designs from participants including gRNAde, the organizers also evaluated the wildtype sequence for each RNA as a sanity check for their assay, as well as two variants of Rosetta's RNA inverse folding protocol (Das et al., 2010).

**OpenKnot score determines success.** For each design, the organizers compute an OpenKnot Score (between 0–100) to measure how likely the sequence is to form the target 3D structure. A score

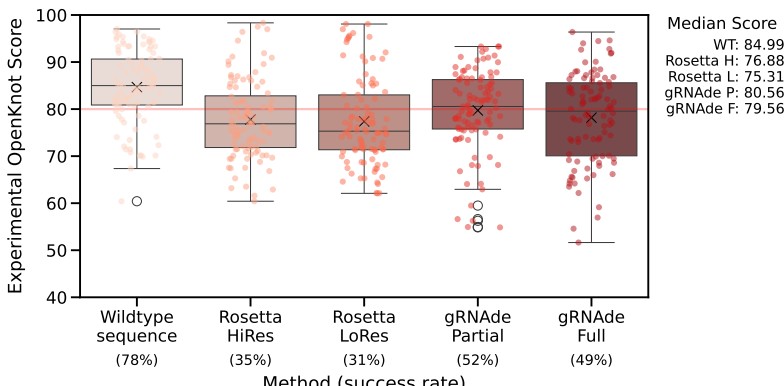

Figure 5: **Wet lab validation of gRNAde designs in Eterna's OpenKnot Round 6.** Each point represents the OpenKnot score (higher is better) for a designed RNA sequence given one of 10 target RNA backbones (10 designs per target per method). Wildtype sequences were included as sanity checks and upper bounds on performance. 50% of gRNAde designs have a score above 80, which is the threshold for a successful design. The median gRNAde design obtains a score of 80, while Rosetta designs have a median score of 76, with a 35% sucess rate. See Appendix Figure 8 for per-puzzle results. Note that we have obtained permission from Eterna organizers to share this data.

above 80 is estimated to be highly likely to form the target structure (scores below 80 may also be successful, but their SHAPE reactivity profiles cannot determine this). The score was developed to aid the discovery and characterisation of pseudoknotted structural elements in RNAs using SHAPE data as part of the OpenKnot challenge series.

**gRNAde has 50% success rate and outperforms Rosetta.** In Figure 5, we plot the distribution of OpenKnot scores across all target RNAs for gRNAde and Rosetta designs as well as the wildtype control sequences. 52% of gRNAde partial designs and 49% of gRNAde full designs obtained OpenKnot scores greater than 80, compared to 35% for the best performing Rosetta variant. The median gRNAde design has a score of 80, while the median Rosetta design scores 76. See Appendix Figure 8 for per-puzzle results where we find that gRNAde designs obtain higher scores than wildtype sequences for 3 targets, suggesting that designed sequences can have higher likelihood of forming the target structure compared to sequences observed in nature.

## 6    CONCLUSION

We introduce gRNAde, a geometric deep learning pipeline for RNA sequence design conditioned on one or more 3D backbone structures. gRNAde represents a significant advance over physics-based Rosetta in terms of both computational and experimental performance, as well as inference speed and ease-of-use. Further, gRNAde enables explicit multi-state design for structurally flexible RNAs which was previously not possible with Rosetta. gRNAde's perplexity correlates with native sequence and structural recovery, and can be used for zero-shot ranking of mutants in RNA engineering campaigns. gRNAde is also the first geometric deep learning architecture for multi-state biomolecule representation learning; the model is generic and can be repurposed for other learning tasks on conformational ensembles, including multi-state protein design.

**Limitations.** Key avenues for future development of gRNAde include supporting multiple interacting chains, accounting for partner molecules with RNAs, and supporting negative design against undesired conformations. We discuss practical tradeoffs to using gRNAde in real-world RNA design scenarios in Appendix E, including limitations due to the current state of 3D RNA structure prediction tools. Finally, we are hopeful that advances in RNA structure determination and computationally assisted cryo-EM (Kappel et al., 2020; Bonilla & Kieft, 2022) will further increase the amount of RNA structures available for training geometric deep learning models in the future.

## ACKNOWLEDGEMENTS

We would like to thank Roger Foo, Phillip Holliger, Alex Borodavka, Rhiju Das, Janusz Bujnicki, Edoardo Gianni, Ben Porebski, Samantha Kwok, Michael Mohsen, Christian Choe, and John Boom for helpful comments and discussions. CKJ was supported by the A*STAR Singapore National Science Scholarship (PhD) and Qualcomm Innovation Fellowship. SVM was supported by the UKRI Centre for Doctoral Training in Application of Artificial Intelligence to the study of Environmental Risks (EP/S022961/1). AM was supported by a U.S. NSF grant (DBI2308699) and two U.S. NIH grants (R01GM093123 and R01GM146340). This research was partially supported by Google TPU Research Cloud and Cambridge Dawn Supercomputer Pioneer Project compute grants.

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

## APPENDICES

# A  RELATED WORK

We attempt to briefly summarise recent developments in RNA structure modelling and design, with an emphasis on deep learning-based approaches.

**RNA inverse folding.**  Most tools for RNA inverse folding focus on secondary structure without considering 3D geometry (Churkin et al., 2018; Runge et al., 2019; Cole et al., 2024) and approach the problem from the lens of energy optimisation (Ward et al., 2023). Rosetta fixed backbone re-design (Das et al., 2010; Leman et al., 2020) is the only energy optimisation-based approach that accounts for 3D structure. Rosetta aims to find the lowest energy RNA sequence for a given structure: (1) Given a starting point such as a target backbone and a random/heuristic starting sequence, Rosetta uses Markov Chain Monte Carlo methods to sample 'moves' which are random changes to the sequence (whether a base is an A, U, G, or C) or side chain rotamer (the orientation of the base). (2) Next, Rosetta updates the all-atom 3D structure of the RNA backbone and uses an all-atom energy function to accept or reject a move based on whether it stabilises the target backbone. This process is repeated until convergence with simulated annealing.

Deep neural networks such as gRNAde can incorporate 3D structural constraints and produce designs in a single forward pass with GPU acceleration, which is orders of magnitude faster than optimisation-based approaches. This is particularly attractive for high-throughput design pipelines as solving the inverse folding optimisation problem is NP hard (Bonnet et al., 2020). Independent of our work, Tan et al. (2023) also developed a contemporaneous deep learning-based 3D RNA inverse folding model.

**RNA structure design.**  Inverse folding models for protein design have often been coupled with backbone generation models which design structural backbones conditioned on various design constraints (Watson et al., 2023; Ingraham et al., 2023; Didi et al., 2023). Current approaches for RNA backbone design use classical (non-learnt) algorithms for aligning 3D RNA motifs (Han et al., 2017; Yesselman et al., 2019), which are small modular pieces of RNA that are believed to fold independently. Such algorithms may be restricted by the use of hand-crafted heuristics and we plan to explore data-driven generative models for RNA backbone design in future work.

**RNA structure prediction.**  There have been several recent efforts to adapt protein folding architectures such as AlphaFold2 (Jumper et al., 2021) and RosettaFold (Baek et al., 2021) for RNA structure prediction (Li et al., 2023b; Wang et al., 2023; Baek et al., 2024). A previous generation of models used GNNs as ranking functions together with Rosetta energy optimisation (Watkins et al., 2020; Townshend et al., 2021). None of these architectures aim at capturing conformational flexibility of RNAs, unlike gRNAde which represents RNAs as multi-state conformational ensembles. Neither can structure prediction tools be used for RNA design tasks as they are not generative models.

**RNA language models.**  Self-supervised language models have been developed for predictive and generative tasks on RNA sequences, including general-purpose models such as RNA FM (Chen et al., 2022) and RiNaLMo (Penic et al., 2024) as well as mRNA-specific CodonBERT (Li et al., 2023a). RNA sequence data repositories are orders of magnitude larger than those for RNA structure (eg. RiNaLMo is trained on 36 million sequences). However, standard language models can only implicitly capture RNA structure and dynamics through sequence co-occurence statistics, which can pose a chellenge for designing structured RNAs such as riboswitches, aptamers, and ribozymes. RibonanzaNet (He et al., 2024) represents a recent effort in developing structure-informed RNA language models by supervised training on experimental readouts from chemical mapping, although RibonanzaNet cannot be used for RNA design. Inverse folding methods like gRNAde are language models conditioned on 3D structure, making them a natural choice for structure-based design.

Table 1: Ablation study and aggregated benchmark results for gRNAde. We report metrics averaged over 100 test sets samples and standard deviations across 3 consistent random seeds. The percentages reported in brackets for the 3D self-consistency scores are the percentage of designed samples within the 'designability' threshold values (scRMSD≤2Å, scTM≥0.45, scGDT≥0.5).

| Split | Max. #states | Model | GNN | Max. train length | Perplexity (↓) | Native seq. recovery (↑) | 2D – EternaFold scMCC (↑) | 3D – RhoFold scRMSD (↓) | scTM-score (↑) | scGDT_TS (↑) |
|---|---|---|---|---|---|---|---|---|---|---|
| Single-state split | 1 | AR | Equiv | 500 | 1.77±0.07 | 0.438±0.01 | 0.624±0.07 | 13.01±1.18 (0.5%) | 0.21±0.0 (14.3%) | 0.22±0.0 (12.7%) |
| | 1 | AR | Equiv | 1000 | 1.73±0.08 | 0.453±0.01 | 0.648±0.01 | 13.10±0.58 (1.0%) | 0.20±0.0 (10.8%) | 0.21±0.0 (10.6%) |
| | 1 | AR | Equiv | 2500 | 1.41±0.01 | 0.513±0.01 | 0.633±0.03 | 11.76±0.91 (1.4%) | 0.27±0.0 (28.8%) | 0.27±0.0 (28.0%) |
| | 1 | AR | Equiv | 5000 | 1.29±0.02 | 0.538±0.03 | 0.612±0.02 | 11.50±0.64 (1.9%) | 0.28±0.0 (32.1%) | 0.28±0.0 (26.2%) |
| | 1 | AR, rand | Equiv | 5000 | 1.59±0.16 | 0.531±0.04 | 0.621±0.04 | 11.87±1.06 (1.9%) | 0.26±0.0 (28.1%) | 0.26±0.0 (24.1%) |
| | 1 | AR | Inv | 5000 | 1.32±0.04 | 0.531±0.01 | 0.585±0.03 | 11.70±0.56 (1.3%) | 0.26±0.0 (24.8%) | 0.25±0.0 (20.1%) |
| | 1 | NAR | Inv | 5000 | 1.54±0.04 | 0.571±0.00 | 0.430±0.02 | 14.26±0.51 (1.3%) | 0.19±0.0 (15.9%) | 0.18±0.0 (12.7%) |
| | 1 | NAR | Equiv | 5000 | 1.46±0.06 | 0.584±0.00 | 0.473±0.02 | 13.04±0.88 (1.3%) | 0.23±0.0 (24.0%) | 0.22±0.0 (17.9%) |
| | 3 | AR | Equiv, DS | 5000 | 1.23±0.05 | 0.539±0.01 | 0.620±0.01 | 11.47±1.05 (2.5%) | 0.28±0.0 (31.4%) | 0.28±0.0 (27.2%) |
| | 5 | AR | Equiv, DS | 5000 | 1.25±0.01 | 0.539±0.02 | 0.596±0.03 | 11.90±1.00 (2.9%) | 0.27±0.0 (31.6%) | 0.26±0.0 (26.4%) |
| | Groundtruth sequence prediction baseline: | | | - | | 1.000±0.00 | 0.686±0.00 | 5.23±0.07 (27.9%) | 0.56±0.0 (68.7%) | 0.55±0.0 (68.7%) |
| | Random sequence prediction baseline: | | | - | | 0.251±0.00 | 0.012±0.00 | 24.40±0.34 (0.0%) | 0.04±0.0 (0.0%) | 0.02±0.0 (0.0%) |
| | ViennaRNA 2D-only baseline: | | | - | | 0.259±0.00 | 0.611±0.00 | 20.34±0.10 (0.0%) | 0.07±0.0 (0.6%) | 0.07±0.0 (1.1%) |
| Multi-state split | 1 | AR | Equiv | 5000 | 1.51±0.01 | 0.481±0.00 | 0.573±0.04 | 21.83±0.53 (0.0%) | 0.12±0.0 (2.6%) | 0.15±0.0 (5.5%) |
| | 3 | AR | Equiv, DS | 500 | 1.87±0.04 | 0.444±0.01 | 0.587±0.02 | 22.09±0.13 (0.0%) | 0.12±0.0 (2.3%) | 0.14±0.0 (5.7%) |
| | 3 | AR | Equiv, DS | 1000 | 1.76±0.04 | 0.455±0.03 | 0.504±0.04 | 22.92±1.43 (0.0%) | 0.11±0.0 (2.3%) | 0.14±0.0 (5.8%) |
| | 3 | AR | Equiv, DS | 2500 | 1.54±0.07 | 0.500±0.01 | 0.543±0.01 | 22.00±0.26 (0.0%) | 0.11±0.0 (2.9%) | 0.14±0.0 (3.7%) |
| | 3 | AR | Equiv, DS | 5000 | 1.44±0.04 | 0.531±0.00 | 0.573±0.03 | 22.19±0.28 (0.0%) | 0.12±0.0 (4.2%) | 0.15±0.0 (7.5%) |
| | 3 | AR | Equiv, DSS | 5000 | 1.37±0.04 | 0.540±0.03 | 0.574±0.03 | 22.20±0.43 (0.0%) | 0.12±0.0 (4.0%) | 0.15±0.0 (7.5%) |
| | 5 | AR | Equiv, DS | 5000 | 1.37±0.03 | 0.510±0.00 | 0.514±0.00 | 21.80±0.08 (0.0%) | 0.12±0.0 (2.9%) | 0.14±0.0 (6.2%) |
| | 1 | NAR | Equiv | 5000 | 1.81±0.03 | 0.489±0.00 | 0.372±0.03 | 24.18±0.63 (0.0%) | 0.09±0.0 (2.2%) | 0.12±0.0 (4.7%) |
| | 3 | NAR | Equiv, DS | 5000 | 1.65±0.13 | 0.506±0.01 | 0.346±0.02 | 24.06±0.43 (0.0%) | 0.08±0.0 (2.0%) | 0.11±0.0 (2.9%) |
| | 3 | NAR | Equiv, DSS | 5000 | 1.60±0.10 | 0.520±0.02 | 0.352±0.03 | 24.18±0.55 (0.0%) | 0.09±0.0 (2.2%) | 0.12±0.0 (4.7%) |
| | 5 | NAR | Equiv, DS | 5000 | 1.59±0.21 | 0.517±0.01 | 0.339±0.01 | 24.16±0.75 (0.0%) | 0.08±0.0 (2.2%) | 0.10±0.0 (4.5%) |
| | Groundtruth sequence prediction baseline: | | | - | | 1.000±0.00 | 0.525±0.00 | 17.52±0.32 (3.9%) | 0.25±0.0 (24.2%) | 0.29±0.0 (31.4%) |
| | Random sequence prediction baseline: | | | - | | 0.249±0.00 | 0.013±0.00 | 31.00±0.20 (0.0%) | 0.03±0.0 (0.0%) | 0.02±0.0 (0.0%) |
| | ViennaRNA 2D-only baseline: | | | - | | 0.258±0.00 | 0.470±0.00 | 29.10±0.00 (0.0%) | 0.05±0.0 (0.0%) | 0.05±0.0 (0.0%) |

# B  ABLATION STUDY

Table 1 presents an ablation study as well as aggregated benchmark for various configurations of gRNAde. Key takeaways are highlighted below. Note that all results in the main paper are reported for models trained on the maximum length of 5000 nucleotides using autoregressive decoding and rotation-equivariant GNN layers, as this lead to the lowest perplexity values.

**Split.** Single- and multi-state splits are described in Section 3; the multi-state split is relatively harder than the single-state split based on overall reduced performance for all baselines and models. The multi-state split evaluates a particularly challenging o.o.d. scenario as the RNAs in the test set have significantly higher structural flexibility compared to those in the training set.

**Max. #states** We evaluate the impact of increasing the maximum number of states as input to gRNAde. Multi-state models improve native sequence recovery as well as structural self-consistency scores over an equivalent single state variant. Notably, on the more challenging multi-state split, the improvement in sequence recovery was observed to be as high as 5-6% for the best multi-state models. This trend holds even for the single-state benchmark where the multi-state model is being used with only one state as input. This suggests that seeing multiple states during training can be useful for teaching gRNAde about RNA conformational flexibility and improve performance even for single-state design tasks.

**GNN and pooling architecture** We ablated whether the internal representations of the GVP-GNN are rotation invariant or equivariant. Equivariant GNNs are theoretically more expressive (Joshi et al., 2023) and we find them more capable at fitting the training distribution (as shown by lower perplexity) which in turn results in improved metrics compared to invariant GNNs.

In order to study the expressivity of pooling in the multi-state setting, we ablate the set pooling function used in the multi-state GNN: Deep Set pooling (DS) as well as the more expressive Deep Symmetric Set pooling (DSS, Maron et al. (2020)). DS pooling is described in Equation (3). In DSS pooling, after *each* encoder GNN layer, we (1) aggregate node embeddings from each single state representation into a pooled multi-state representation per node, (2) apply a Geometric Vector

Perceptron update on the multi-state representation, and (3) add the updated multi-state representation back to each single state node representation. DSS pooling marginally improves performance on out-of-distribution test sets for both the single- and multi-state splits. We notice that DSS models fit the training data significantly better (final training loss goes from 0.40 to 0.36 for 3 states). While DSS pooling is more expressive that DS pooling, it also adds 200K more parameters to the model and doubles training iteration time (4 mins to 8 mins for 3 states).

**Model and decoder**  'AR' implies autoregressive decoding (described in Section 2.2, uses 4 encoder and 4 decoder layers), while 'NAR' implies non-autoregressive, one-shot decoding using an MLP (uses 8 encoder layers). Across both evaluation splits, AR models show significantly higher self-consistency scores than NAR, even though NAR lead to higher sequence recovery for the single-state split. AR is more expressive and can condition predictions at each decoding step on past predictions, while one-shot NAR samples from independent probability distributions for each nucleotide. Thus, AR is a better inductive bias for predicting base pairing and base stacking interactions that are drivers of RNA structure (Vicens & Kieft, 2022). For instance, G-C and A-U pairs can often be swapped for one another, but non-autoregressive decoding does not capture such paired constraints.

Additionally, we also present results for the impact of training gRNAde with random decoding order. This can be practically very useful for partial or conditional design scenarios, and leads to a minor reduction in sequence recovery and 3D self-consistency (in line with what was observed for ProteinMPNN).

**Max. train RNA length**  Limiting the maximum length of RNAs used for training can be seen as ablating the use of ribosomal RNA families (which are thousands of nucleotides long and form complexes with specialised ribosomal proteins). We find that training on only short RNAs fewer than 1000s of nucleotides leads to worse sequence recovery and 3D self-consistency scores, even though it improves 2D self-consistency across both evaluation splits. This suggests that tertiary interactions learnt from ribosomal RNAs can generalise to other RNA families to some extent (large ribosomal RNAs were excluded from test sets).

**Non-learnt baselines.**  We report the performance of two non-learnt baselines to contextualise gRNAde's performance: for each test sample, simply predicting the groundtruth sequence back and predicting a random sequence. Structural self-consistency scores for the Groundtruth baseline provides a rough upper bounds on the maximum score that any gRNAde designs can theoretically obtain given the current state of 2D/3D structure predictors being used. gRNAde always performs better than the random baseline and often reaches 2D self-consistency scores close to the upper bound. Both 2D and 3D self-consistency scores are inherently limited by the performance of the structure prediction methods used.

**2D inverse folding baseline.**  We additionally report results for ViennaRNA's 2D-only inverse folding method to further demonstrate the utility of 3D inverse folding. ViennaRNA has improved 2D self-consistency scores over gRNAde but fails to capture tertiary interactions in its designs, as evident by poor recovery and 3D self-consistency scores similar to the random baseline. We observed the same trend for other 2D-only inverse folding methods such as NuPack's design tool. This result should not be surprising, as 2D tools are meant for design scenarios that only involve base pairing and do not take any 3D information into account.

**Choice of structure predictors.**  As previously noted, self-consistency metrics are highly dependent on the performance of the structure prediction method used. We chose EternaFold as it is simple to use as well as validated for *designed* and synthetic RNAs, unlike most other 2D structure prediction tools. Replacing EternaFold with RNAFold lead to unchanged results and did not modify the relative rankings of the models:

- AR, 1 state, Equiv. GNN, EternaFold scMCC: $0.612 \pm 0.02$, RNAFold scMCC: $0.614 \pm 0.03$.
- NAR, 1 state, Equiv. GNN, EternaFold scMCC: $0.473 \pm 0.02$, RNAFold scMCC: $0.477 \pm 0.04$.

Lastly, we would like to note the challenge of evaluating multi-state design: Structural self-consistency metrics are not ideal for evaluating RNAs which do not have one fixed structure/undergo changes to their structure. It would be ideal (but extremely slow and expensive) to run MD simulations to validate multi-state design models.

# C    ADDITIONAL RESULTS

Table 2: Full results for Figure 2 comparing gRNAde to Rosetta, FARNA and ViennaRNA for single-state design on 14 RNA structures of interest identified by Das et al. (2010). Rosetta and FARNA recovery values are taken from Das et al. (2010), Supplementary Table 2.

| PDB ID | Description | ViennaRNA Recovery | FARNA Recovery | RDesign Recovery | Rosetta Recovery | gRNAde (single-state) Recovery | Perplexity | 2D self-cons. |
|---|---|---|---|---|---|---|---|---|
| 1CSL | RRE high affinity site | 0.25 | 0.20 | 0.4455 | 0.44 | 0.5719 | 1.2812 | 0.8644 |
| 1ET4 | Vitamin B12 binding RNA aptamer | 0.25 | 0.34 | 0.3929 | 0.44 | 0.6250 | 1.3457 | -0.0135 |
| 1F27 | Biotin-binding RNA pseudoknot | 0.30 | 0.36 | 0.3013 | 0.37 | 0.3437 | 1.6203 | 0.4523 |
| 1L2X | Viral RNA pseudoknot | 0.24 | 0.45 | 0.3727 | 0.48 | 0.4721 | 1.3181 | 0.5692 |
| 1LNT | RNA internal loop of SRP | 0.33 | 0.27 | 0.5556 | 0.53 | 0.5843 | 1.4337 | 0.1379 |
| 1Q9A | Sarcin/ricin domain from E.coli 23S rRNA | 0.27 | 0.40 | 0.4417 | 0.41 | 0.5044 | 1.3411 | 0.0597 |
| 4FE5 | Guanine riboswitch aptamer | 0.29 | 0.28 | 0.4112 | 0.36 | 0.5300 | 1.3824 | 0.9116 |
| 1X9C | All-RNA hairpin ribozyme | 0.26 | 0.31 | 0.3967 | 0.50 | 0.5000 | 1.3905 | 0.6630 |
| 1XPE | HIV-1 B RNA dimerization initiation site | 0.27 | 0.24 | 0.3834 | 0.40 | 0.7037 | 1.2177 | 0.7768 |
| 2GCS | Pre-cleavage state of glmS ribozyme | 0.25 | 0.26 | 0.4518 | 0.44 | 0.5078 | 1.3053 | 0.4062 |
| 2GDI | Thiamine pyrophosphate-specific riboswitch | 0.25 | 0.38 | 0.3523 | 0.48 | 0.6500 | 1.2363 | -0.0251 |
| 2OEU | Junctionless hairpin ribozyme | 0.23 | 0.30 | 0.5000 | 0.37 | 0.9519 | 1.0913 | 0.7768 |
| 2R8S | Tetrahymena ribozyme P4-P6 domain | 0.27 | 0.36 | 0.5641 | 0.53 | 0.5689 | 1.1881 | 0.7281 |
| 354D | Loop E from E. coli 5S rRNA | 0.28 | 0.35 | 0.4458 | 0.55 | 0.4410 | 1.4938 | 0.0430 |
|  | Overall recovery: | 0.27 | 0.32 | 0.4296 | 0.45 | 0.5682 |  |  |

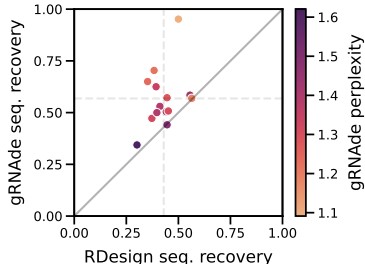

Figure 6: gRNAde compared to RDesign for single-state design.

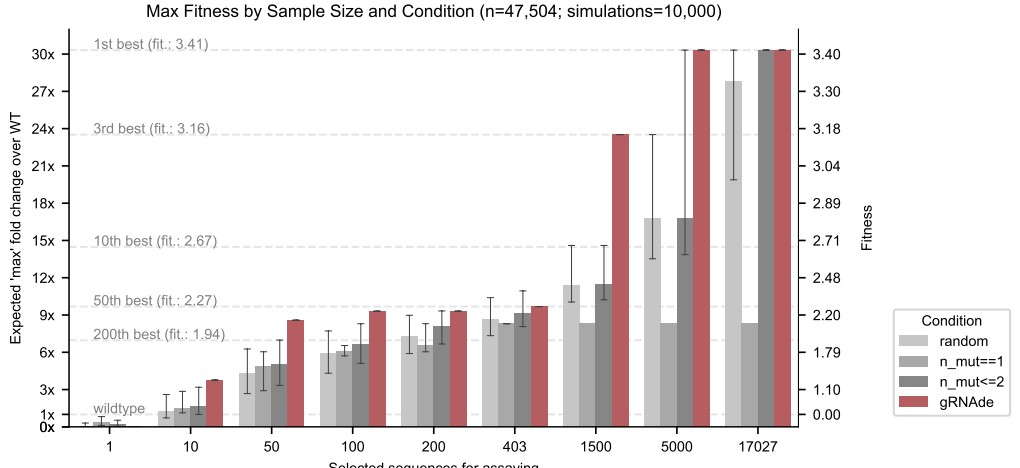

Figure 7: **Retrospective study of gRNAde for ranking ribozyme mutant fitness (t1 subunit).** Using the backbone structure and mutational fitness landscape data from an RNA polymerase ribozyme (McRae et al., 2024), we retrospectively analyse how well we can rank variants at multiple design budgets using random selection vs. gRNAde's perplexity for mutant sequences conditioned on the backbone structure (scaffolding subunit t1). gRNAde performs better than single site saturation mutagenesis, even when all single mutants are explored (total of 403 single mutants, 17,027 double mutants for the scaffolding subunit t1 in McRae et al. (2024)). See Section 4.3 for results on catalytic subunit 5TU and further discussions.

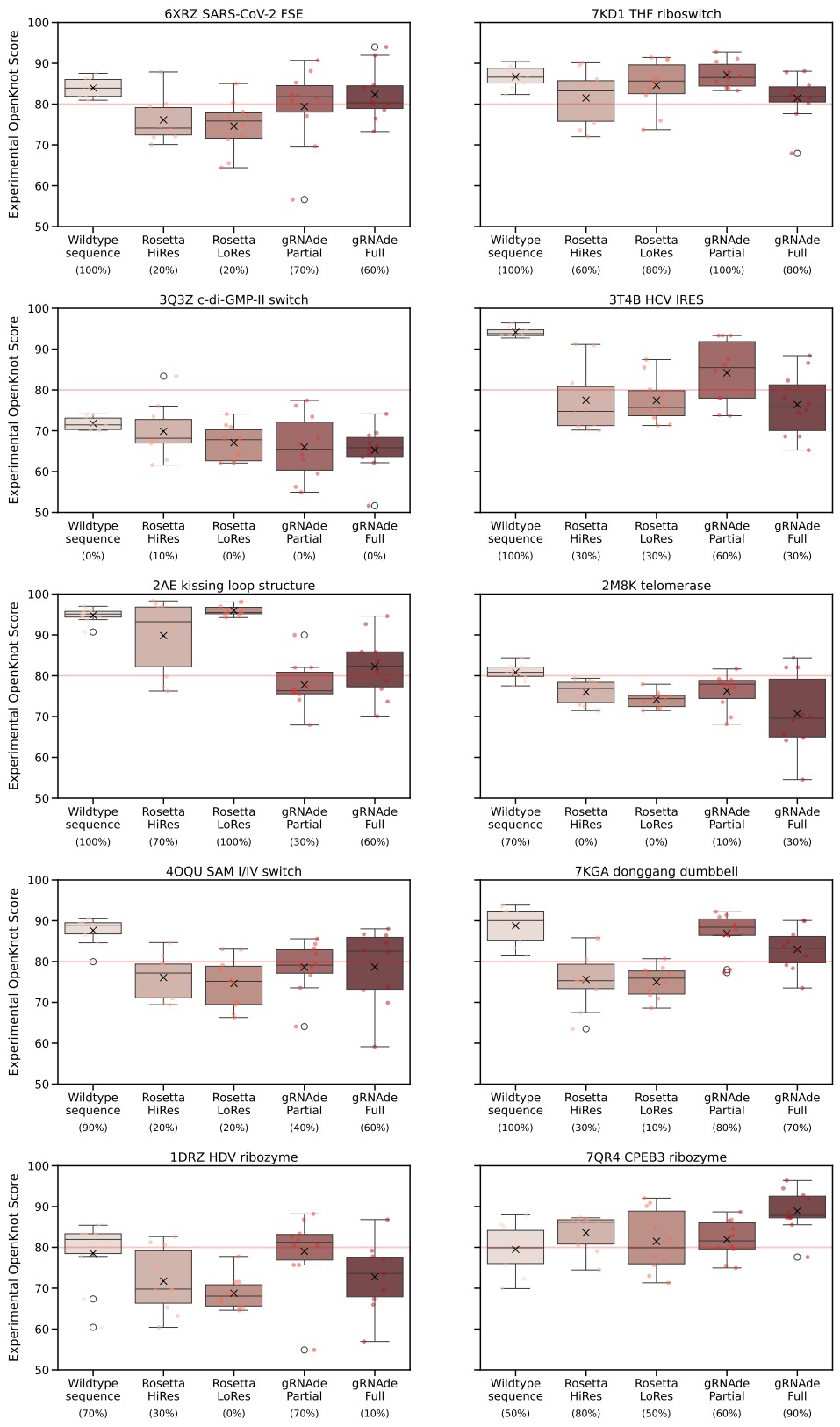

Figure 8: **Per-input wet lab validation results of gRNAde designs.** Notably, for the HDV ribozyme, CPEB3 ribozyme and donggang dumbell backbone structures, gRNAde designs tend to obtain higher OpenKnot scores than the wildtype sequences, which suggests that designed sequences can have higher likelihood of forming the pseudoknotted structure compared to naturally observed sequences.

# D    ADDITIONAL FIGURES

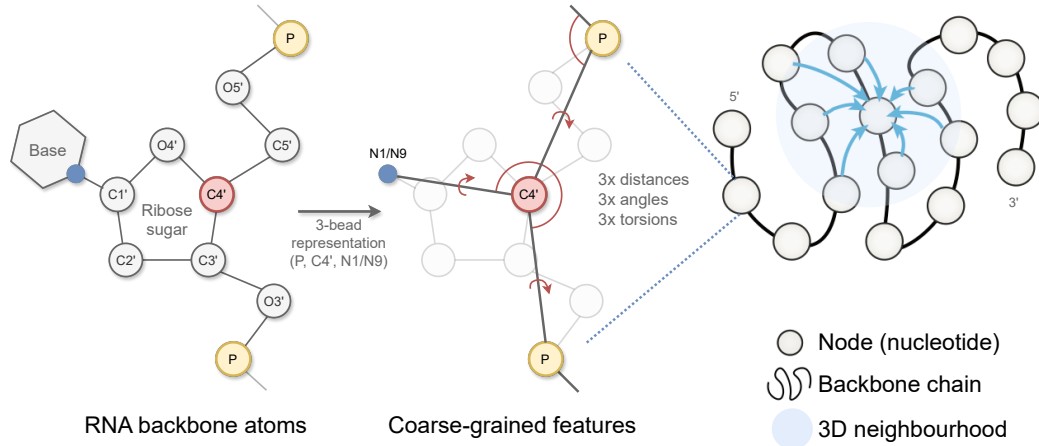

Figure 9: **gRNAde featurizes RNA backbone structures as 3D geometric graphs.** Each RNA nucleotide is a node in the graph, consisting of 3 coarse-grained beads for the coordinates for P, C4', N1 (pyrimidines) or N9 (purines) which are used to compute initial geometric features and edges to nearest neighbours in 3D space. Backbone chain figure adapted from Ingraham et al. (2019).

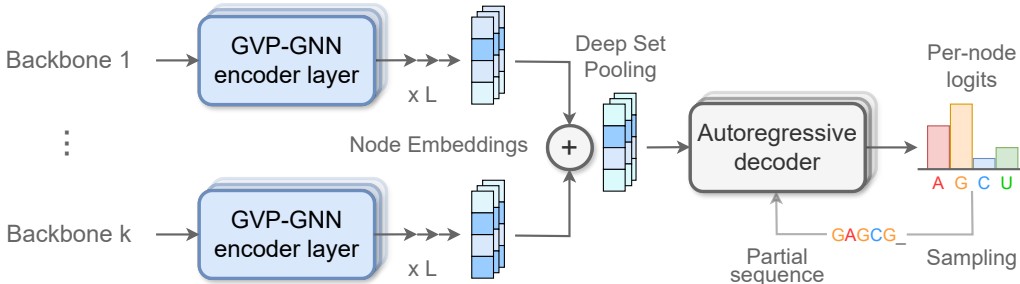

Figure 10: **gRNAde model architecture.** One or more RNA backbone geometric graphs are encoded via a series of SE(3)-equivariant Graph Neural Network layers (Jing et al., 2020) to build latent representations of the local 3D geometric neighbourhood of each nucleotide within each state. Representations from multiple states for each nucleotide are then pooled together via permutation invariant Deep Sets (Zaheer et al., 2017), and fed to an autoregressive decoder to predict a probabilities over the four possible bases (A, G, C, U). The probability distribution can be sampled to design a set of candidate sequences. During training, the model is trained end-to-end by minimising a cross-entropy loss between the predicted probability distribution and the true sequence identity.

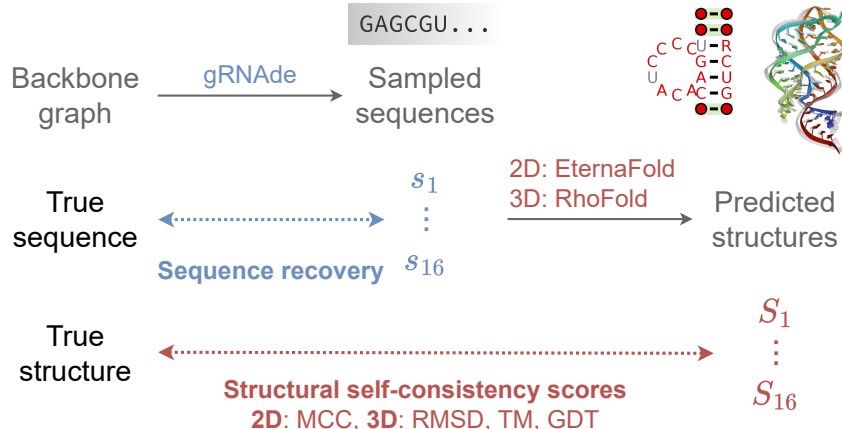

Figure 11: **In-silico evaluation metrics for gRNAde designed sequences.** We consider (1) *sequence recovery*, the percentage of native nucleotides recovered in designed samples, (2) *self-consistency scores*, which are measured by 'forward folding' designed sequences using a structure predictor and measuring how well 2D and 3D structure are recovered (we use EternaFold and RhoFold for 2D/3D structure prediction, respectively). We also report (3) *perplexity*, the model's estimate of the likelihood of a sequence given a backbone.

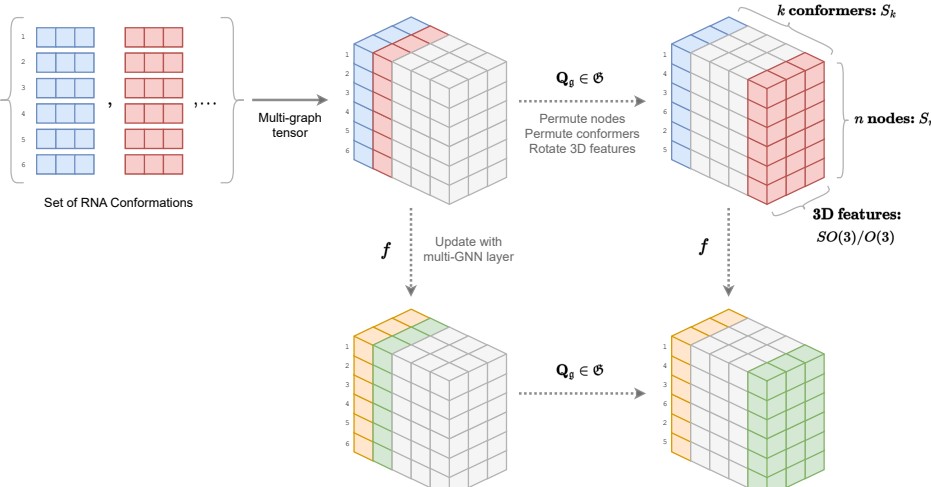

Figure 12: **Multi-graph tensor representation of RNA conformational ensembles**, and the associated symmetry groups acting on each axis. We process a set of $k$ RNA backbone conformations with $n$ nodes each into a tensor representation. Each multi-state GNN layer updates the tensor while being equivariant to the underlying symmetries; pseudocode is available in Listing 1. Here, we show a tensor of 3D vector-type features with shape $n \times k \times 3$. As depicted in the equivariance diagram, the updated tensor must be equivariant to permutation $S_n$ of $n$ nodes for axis 1, permutation $S_k$ of $k$ conformational states for axis 2, and rotation $SO(3)/O(3)$ of the 3D features for axis 3.

```python
class MultiGVPConv(MessagePassing):
    '''GVPConv for handling multiple conformations'''

    def __init__(self, ...):
        ...

    def forward(self, x_s, x_v, edge_index, edge_attr):

        # stack scalar feats along axis 1:
        # [n_nodes, n_conf, d_s] -> [n_nodes, n_conf * d_s]
        x_s = x_s.view(x_s.shape[0], x_s.shape[1] * x_s.shape[2])

        # stack vector feat along axis 1:
        # [n_nodes, n_conf, d_v, 3] -> [n_nodes, n_conf * d_v*3]
        x_v = x_v.view(x_v.shape[0], x_v.shape[1] * x_v.shape[2]*3)

        # message passing and aggregation
        message = self.propagate(
            edge_index, s=x_s, v=x_v, edge_attr=edge_attr)

        # split scalar and vector channels
        return _split_multi(message, d_s, d_v, n_conf)

    def message(self, s_i, v_i, s_j, v_j, edge_attr):

        # unstack scalar feats:
        # [n_nodes, n_conf * d] -> [n_nodes, n_conf, d_s]
        s_i = s_i.view(s_i.shape[0], s_i.shape[1]//d_s, d_s)
        s_j = s_j.view(s_j.shape[0], s_j.shape[1]//d_s, d_s)

        # unstack vector feats:
        # [n_nodes, n_conf * d_v*3] -> [n_nodes, n_conf, d_v, 3]
        v_i = v_i.view(v_i.shape[0], v_i.shape[1]//(d_v*3), d_v, 3)
        v_j = v_j.view(v_j.shape[0], v_j.shape[1]//(d_v*3), d_v, 3)

        # message function for edge j-i
        message = tuple_cat((s_j, v_j), edge_attr, (s_i, v_i))
        message = self.message_func(message)  # GVP

        # merge scalar and vector channels along axis 1
        return _merge_multi(*message)

def _split_multi(x, d_s, d_v, n_conf):
    '''
    Splits a merged representation of (s, v) back into a tuple.
    '''
    s = x[..., :-3 * d_v * n_conf].view(x.shape[0], n_conf, d_s)
    v = x[..., -3 * d_v * n_conf:].view(x.shape[0], n_conf, d_v, 3)
    return s, v

def _merge_multi(s, v):
    '''
    Merges a tuple (s, v) into a single `torch.Tensor`,
    where the vector channels are flattened and
    appended to the scalar channels.
    '''
    # s: [n_nodes, n_conf, d] -> [n_nodes, n_conf * d_s]
    s = s.view(s.shape[0], s.shape[1] * s.shape[2])
    # v: [n_nodes, n_conf, d, 3] -> [n_nodes, n_conf * d_v*3]
    v = v.view(v.shape[0], v.shape[1] * v.shape[2]*3)
    return torch.cat([s, v], -1)
```

Listing 1: **PyG-style pseudocode for a multi-state GVP-GNN layer.** We update node features for each conformational state independently while maintaining permutation equivariance of the updated feature tensors along both the first (no. of nodes) and second (no. of conformations) axes.

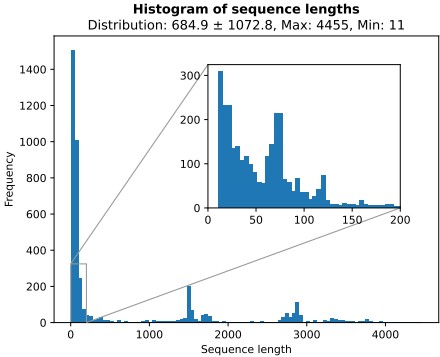

(a) **Sequence length.** The dataset is long-tailed in terms of RNA sequence length, with many short sequences including aptamers, riboswitches, ribozymes, and tRNAs (fewer than 200 nucleotides). The dataset also includes several longer ribosomal RNAs (thousands of nucleotides).

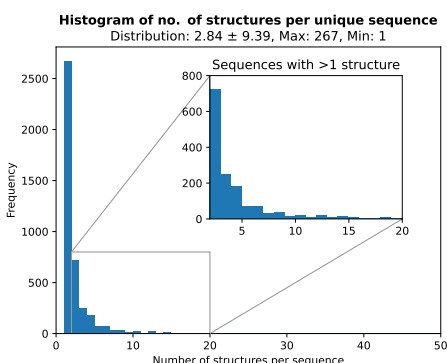

(b) **Number of structures per sequence.** The dataset covers a wide range of RNA conformation ensembles, with on average 3 structures per sequence. There are multiple structures available for 1,547 sequences. The remaining 2,676 sequences have one corresponding structure.

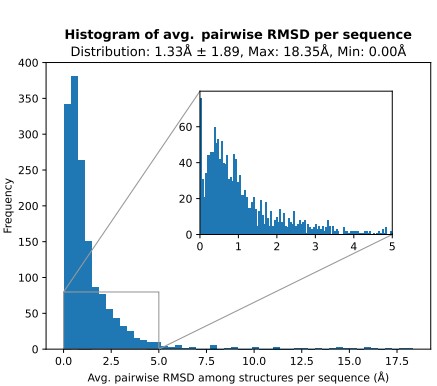

(c) **Average pairwise RMSD per sequence.** For 1,547 sequences with multiple structures, there is significant structural diversity among conformations. On average, the pairwise C4' RMSD among the set of structures for a sequence is greater than 1Å.

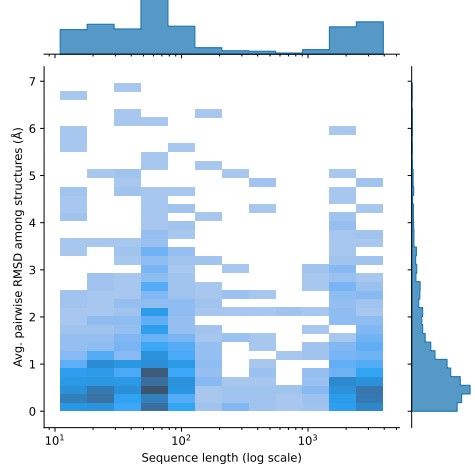

(d) **Bivariate distribution for sequence length vs. avg. RMSD.** The joint plot illustrates how structural diversity (measured by avg. pairwise RMSD) varies across sequence lengths. We notice similar structural variations regardless of sequence length.

Figure 13: **RNASolo data statistics.** We plot histograms to visualise the diversity of RNAs available in terms of (a) sequence length, (b) number of structures available per sequence, as well as (c) structural variation among conformations for those RNA that have multiple structures. The bivariate distribution plot (d) for sequence length vs. average pairwise RMSD illustrates structural diversity regardless of sequence lengths.

# E  FAQS ON USING GRNADE

**How to chose the number of states to provide as input to gRNAde?**  In general, this would depend on the design objective. For instance, designing riboswitches may necessitate multi-state design, while a single-state pipeline may be more sensible for locking an aptamer into its bound conformation (Yesselman et al., 2019). Note that it may be possible to benefit from multi-state gRNAde models even when performing single-state design by using slightly noised variations of the same backbone structure as an input conformational ensemble.

**How to prioritise or chose amongst designed sequences?**  We have currently provided 3 types of evaluation metrics: native sequence recovery, structural self-consistency scores and perplexity, towards this end. We suspect that recovery may not be the ideal choice, except for design scenarios where we require certain regions of the RNA sequence to be conserved or native-like. Self-consistency scores may provide an overall more holistic evaluation metric as they accounts for alternative base pairings which still lead to similar structures as well as better capture the recovery of structural motifs responsible for functionality. However, structural self-consistency scores inherit the limitations of the structure prediction methods used as part of their computation. For instance, computing the self-consistency score between an RNA backbone and its own native sequence provides an upper bounds on the maximum score that designs can obtain under a given structure prediction method. Lastly, gRNAde's perplexity estimates the likelihood of a sequence given a backbone and can be useful for ranking designs and mutants in RNA engineering campaigns (especially for design scenarios where structure prediction tools are not performant).

In real-world design scenarios, we can pair gRNAde with another machine learning model (an 'oracle') for ranking or predicting the suitability of designed sequences for the objective (for instance, binding affinity or some other notion of fitness). We hope to conduct further experimental validation of gRNAde designs in the wet lab in order to better understand these tradeoffs.

**Why not average single-state logits over multiple states for multi-state design?** ProteinMPNN (Dauparas et al., 2022) proposes to average logits from multiple backbones for multi-state protein design. Here is a simple example to highlight issues with such an approach: Consider two states A and B, and choice of labels X, Y, and Z. For state A: X, Y, Z are assigned probabilities 75%, 20%, 5%. For state B: X, Y, Z are assigned probabilities 5%, 20%, 75%. Logically, label Y is the only one that is compatible with both states. However, averaging the probabilities would lead to label X or Z being more likely to be sampled in designs. As an alternative, gRNAde is based on multi-state GNNs which can take as input one or more backbone structures and generate sequences conditioned on the conformational ensemble directly.

## F    COMPARISON TO CONTEMPORANEOUS WORK

Independently of our work, Tan et al. (2023) also developed RDesign, a deep learning-based 3D RNA inverse folding model. While these papers were developed concurrently in 2023, RDesign was first to be accepted for formal publication at ICLR 2024. Here, we want to highlight key difference between gRNAde and RDesign, as well as technical/reproducibility concerns with Tan et al. (2023).

- **Experimental validation**: gRNAde designs work in the wet lab and have a significantly higher experimental success rate than Rosetta. RDesign's methodology restricts practical applicability for real-world usage (see subsequent points on sampling).

- **Methodology differences and issues with RDesign**:
  - *New capabilities*: gRNAde enables explicit multi-state design to generate sequences conditioned on multiple backbone structures, which is not possible with Rosetta nor RDesign. We have also demonstrated the utility of gRNAde's perplexity for zero-shot ranking of mutants in RNA engineering campaigns.
  - *Decoding*: gRNAde uses an autoregressive decoder with rotation-equivariant GNN layers, while RDesign uses a non-autoregressive (one-shot) decoder with rotation-invariant layers. In our ablation study in Appendix B, we found autoregressive decoding to show significantly higher 2D and 3D self-consistency scores than non-autoregressive decoding, even though non-autoregressive decoding lead to higher sequence recovery. We also found that equivariant GNN layers improve performance over invariant layers.
  - *Sampling*: As of 01/08/2024, RDesign does *not* implement any sampling operation during inference, despite having a method called 'sample' in its model class. This means that any design produced by RDesign is deterministic and not diverse, restricting its practical usage. This also means that the metric reported as 'recovery' in the RDesign paper does not follow the standard definition in the protein design community. (Typically, computing recovery requires drawing samples from the probability distribution learnt by the model, whereas the RDesign paper reports classification accuracy.)

- **Evaluation and data splitting differences**:
  - *Evaluation metrics*: RDesign's evaluation measures native sequence recovery, only. We have additionally introduced structural self-consistency metrics at the 2D and 3D level, which have been shown to better correlate with experimental success in protein design.
  - *Perplexity*: We found gRNAde's perplexity to be correlated with sequence and structural recovery, as well as demonstrated its utility for zero-shot ranking of mutants in RNA engineering. On the other hand, RDesign does not report perplexity and claims that perplexity is an unsuitable metric for RNA design. RDesign directly outputting probability distributions and not using any sampling (see previous point) can be one possible explanation of why RDesign models produce unsual perplexity values (Appendix E.3 of their paper, as well as looking through the training logs released for their best checkpoint: training perplexity is close to 1, while validation perplexity is more than 22, suggesting overfitting and extremely poor generalisation).
  - *Data splitting*: While both studies use structural clustering to evaluate generalisation to structurally dissimilar RNAs, RDesign's test splits are determined randomly without justification of whether the RNAs used for the test set are scientifically relevant. Our experiments use an expert curated test split of high-quality structured RNAs from Das et al. (2010) to fairly compare gRNAde to Rosetta, as well as a new split based on conformational flexibility to benchmark multi-state design.

- **Usage and reproducibility issues with RDesign**:
  - We release open source training and inference code as well as model checkpoints to enable complete reproducibility. We also release Colab notebooks and detailed tutorials to make gRNAde broadly applicable and useful in real-world RNA design campaigns.
  - As of 01/08/2024, RDesign's codebase did not provide any installation instructions or training code to reproduce their work. None of the model, dataset, and trainer classes had any documentation regarding the expected data format, expected inputs and their shapes, etc. The dataset files released by RDesign were found to be corrupted (we tried loading them on a Macbook and two different Linux servers), all of which produced a `RuntimeError` stating "Invalid magic number; corrupt file".

- **Improved performance of gRNAde over RDesign**:
  - As of 01/08/2024, despite the reproducibility issues and lack of documentation in the code, we were able to reverse-engineer the best model checkpoint released with RDesign and run inference on our single-state evaluation set. In Figure 2a and Table 2, we find that RDesign significantly underperforms gRNAde as well as Rosetta, obtaining an overall recovery rate of 43% compared to 45% for Rosetta and 56% for gRNAde. In Figure 6, we see that gRNAde outperforms RDesign on all 14 of the high-quality structured RNAs of interest identified by Das et al. (2010).
  - In our ablation studies in Appendix B, we fairly compare the performance of gRNAde's autoregressive decoder and equivariant GNN layers with non-autoregressive and invariant GNN variants (which are what RDesign uses in their architecture). This is a direct, apples-to-apples comparison of key architectural differences between gRNAde and RDesign in order to uncover the source of improvement. We find that gRNAde variants with non-autoregressive decoding can improve sequence recovery but that autoregressive decoding has significantly higher 2D and 3D self-consistency scores (which we care about more in real-world design scenarios). We also find that equivariant GNN layers improve performance over invariant GNN layers.

We believe our study brings significant new contributions and insights as well as resources to the community, and that there is space for multiple papers offering different perspectives on the same topic. At the same time, we have found it challenging to reproduce Tan et al. (2023) due to several methodology and reproducibility issues, which we want to highlight via this section.

