# OpenReview forum: "gRNAde: Geometric Deep Learning for 3D RNA inverse design"
_ICLR.cc/2025/Conference — ICLR 2025 Spotlight_

### Official Review · Reviewer_sLwi · 2024-10-31

**Soundness:** 3
**Presentation:** 3
**Contribution:** 3
**Rating:** 6
**Confidence:** 5

**Summary:**

This paper is dedicated to solving the inverse folding problem of RNA. Given multiple 3D conformations of an RNA, the method designs the corresponding RNA sequence. The experimental section demonstrates the effectiveness of this approach.

**Strengths:**

This is an interesting paper. The biggest difference from past methods is that this paper can take into account the flexibility of RNA structures, accept conformations of multiple RNAs simultaneously, and designs a reasonable model. I agree that this is a valid innovation point. This inclines me to accept the paper.

**Weaknesses:**

Although this paper contains some very interesting innovations, several key issues have led me to give it the current rating. I hope the authors can fix these problems. If these issues can be resolved, I will consider improving my rating.

1. In the first sentence of the abstract, the authors claim: "Computational RNA design tasks are often posed as inverse problems, where sequences are designed based on adopting a single desired secondary structure $\textbf{without considering 3D geometry}$ and conformational diversity." I believe that the phrase "without considering 3D geometry" is factually incorrect. Because RDesign[1] performs inverse folding based on the 3D structure of RNA, it should take 3D geometry into account. The authors should provide a reasonable explanation or correct all descriptions in the paper that claim "without considering 3D geometry."

[1] RDESIGN: HIERARCHICAL DATA-EFFICIENT REPRESENTATION LEARNING FOR TERTIARY STRUCTURE-BASED RNA DESIGN. $\textbf{ICLR 2024}$

2. The experimental section lacks a comparison with RDESIGN, which is also deep-learning-based. Moreover, in the abstract, gRNAde should be compared with RDESIGN rather than Rosetta, which would be more reasonable. Figure 3(a) shows the performance of gRNAde under different numbers of conformations. This experiment also needs to be compared with RDESIGN. Although RDESIGN can only accept one conformation, the authors should use the result of RDESIGN with a single conformation as a baseline for comparison.

**Questions:**

None

---

> ### Author Response · Authors · 2024-11-15
> **Thank you for your positive comments and feedback on presentation**
>
> > ... The authors should provide a reasonable explanation or correct all descriptions in the paper that claim "without considering 3D geometry."
>
> The first line of the abstract is more of a **stylistic choice** than a technical assertion. Our intention is to illustrate that *most* research on RNA inverse design has focussed on 2D structure except Rosetta (2010), until recent works like RDesign and gRNAde revisited the task from the 3D perspective. This review supports our sentence: https://academic.oup.com/bib/article/19/2/350/2666340 (all the methods they benchmark are 2D-only)
>
> We have not made the claim that other methods do not consider 3D geometry anywhere else in the paper or that we are the first to do so.
>
>
> > In the abstract, gRNAde should be compared with RDESIGN rather than Rosetta, which would be more reasonable.
>
> We found Rosetta to outperform RDesign in the single-state benchmark of 3D RNAs of interest identified by Das et al. (2010) in Figure 2, so we thought it was reasonable to state metrics w.r.t. the next best performing method.
>
> Rosetta is also very well known in the broader biomolecule design community, so we think this will help readers better contextualize our work if they only skim the abstract.
>
>
> > Figure 3(a) shows the performance of gRNAde under different numbers of conformations. This experiment also needs to be compared with RDESIGN. Although RDESIGN can only accept one conformation, the authors should use the result of RDESIGN with a single conformation as a baseline for comparison.
>
> Thanks for the suggestion - we have added the result for RDesign into the revised manuscript Figure 3. We found that gRNAde (0.48-0.53) outperforms RDesign (0.38) in the multi-state design benchmark, too.
>
>
> > The experimental section lacks a comparison with RDESIGN, which is also deep-learning-based.
>
> We have directly compared gRNAde to Rdesign. See Figure 2a, Appendix Table 2, as well as Figure 6. We also added RDesign results to Figure 3 as per your suggestion. Overall, we found that gRNAde outperforms RDesign.

---

> ### Comment · Reviewer_sLwi · 2024-11-15
>
> Because the author added a comparison with RDesign, I increased my score.
>
> However, regarding the issue of "without considering 3D geometry," I still believe it has not been resolved. First, I acknowledge that the author's statement, "We have not made the claim that other methods do not consider 3D geometry anywhere else in the paper or that we are the first to do so," is correct. Strictly speaking, this may not be considered a real-time error. However, my points are as follows:
>
> 1. Writing in this way in the abstract is misleading and can lead readers to believe that this paper aims to address the issue of past methods not considering 3D geometry. Since this paper is not the first to solve this problem, I think it is inappropriate to write it this way.
> 2. I do not consider considering 3D geometry to be an interesting innovation, and the negative impact of writing it this way outweighs the positive impact. Even ignoring RDesign, inverse folding based on 3D structures has already been a very classic method in proteins, such as ProteinMPNN. It is clear that such methods can be transferred to RNA, so I do not think considering RNA 3D geometry is an interesting innovation. What truly impressed me about this paper is the idea of considering multiple conformations, which is the main content. The authors should emphasize the idea of considering multiple conformations more fully. However, in the abstract, the authors also emphasize considering RNA 3D geometry, which distracts the reader's attention. It shifts the reader's focus from a very interesting idea (multiple conformations) to a relatively uninteresting point (3D geometry). Therefore, I believe it is a loss.

---

> > ### Author Response · Authors · 2024-11-15
> >
> > Thanks for your suggestion. We updated the abstract to state the following in order to change the emphasis:
> >
> > "...sequences are designed based on adopting a single desired secondary structure without considering 3D conformational diversity"
> >
> > It is highlighted in blue in the revised manuscript

---

> > > ### Comment · Reviewer_sLwi · 2024-11-16
> > >
> > > that’s good.

---

> > > > ### Author Response · Authors · 2024-11-19
> > > >
> > > > Thanks for acknowledging -- would you be willing to update your score for Presentation from "1: poor" if you are happy with how we have addressed your presentation-related suggestions?

---

> > > > > ### Comment · Reviewer_sLwi · 2024-11-19
> > > > >
> > > > > Done

---

> > > > > > ### Author Response · Authors · 2024-11-29
> > > > > >
> > > > > > You’ve rated our paper - Soundness: 4: excellent, Presentation: 3: good, Contribution: 3: good - but with an overall rating 6: marginally above acceptance. It sounds like you rate our work as a good paper, overall, based on the 3 criteria.
> > > > > >
> > > > > > Is there something we can do in this extended discussion period to improve your score?

---

### Official Review · Reviewer_47od · 2024-11-01

**Soundness:** 3
**Presentation:** 3
**Contribution:** 2
**Rating:** 8
**Confidence:** 3

**Summary:**

This paper presents gRNAde, a geometric deep learning-based approach for RNA inverse design task focused on creating RNA sequences conditioned on provided 3D conformations. gRNAde supports single- and multi-state RNA designs. The method has been compared against physics based tools such as Rosetta software with gRNAde showing better performance in multiple applications with faster processing times. The paper also shows gRNAde's potential for zero-shot learning in RNA fitness landscapes.

**Strengths:**

1. The paper studies an important problem of RNA inverse design inspired by similar works in protein space. Given that RNA 3D experimental data is much more limited and the structure of RNA being much more flexible than proteins, innovative approaches for this problem are desirable. The method although builds on existing architectures, employs sensible design suitable for RNA modeling (e.g.- using a three-bead (pseudotorsional) representation of the RNA backbone, minimizing the high-dimensional torsional space).
2. gRNAde introduces an innovative multi-state GNN that models conformational ensembles—distinct structural states of RNA—by encoding them as a geometric multi-graph. The model's ability to simultaneously consider multiple conformations (or states) enhances its accuracy in predicting sequences that can adopt different functional forms.
3. Benchmarking against Rosetta and other tools shows higher higher sequence recovery rate and speedup for gRNAde showcasing its usability for for high-throughput scenarios.
4. The paper is well-written and easy to follow.

**Weaknesses:**

Although the paper explores an interesting problem and presents a sensible approach, there are some aspects which can be improved.


1. I did not find any discussion on if the model explicitly incorporates any biological nuances such as RNA modifications, cellular localization, or folding kinetics, which are essential for in vivo RNA functionality. Given that experimental training data is limited for RNA, such information may enhance performance especially that Physics based models often use such information as well. Some discussion around this can be helpful.

2. The evaluation focuses primarily on sequence recovery and perplexity. However, there are other aspects such as structural stability, functional motifs, etc. that affect RNA design which if not accounted for may produce RNA designs that are theoretically correct but biologically unviable. Did the authors perform any analysis of these aspects in their designed RNAs?

3. There are various ways to extract or represent the RNA backbone in molecular modeling and structural biology without clear consensus (see Richardson et al., 2008, RNA backbone: Consensus all-angle conformers and modular string nomenclature). Did the authors experiment with other backbone representations to see the impact of the particular choice of backbone representations?

4. From evaluation perspective, I have several suggestions which may help understanding the performance of gRNAde:
 (a) I did not see any analysis on failure cases, thus limiting the understanding of scenarios where gRNAde may perform poorly, hindering its reliability and robustness.
(b) Additionally, an evaluation of how varying RNA sequence lengths affect the performance may lead to better understanding of the pros and cons of gRNAde.
(c) Finally, pseudoknots are important in RNA secondary structure, which can lead to incomplete or inaccurate folding predictions, as pseudoknots play a critical role in the functional conformation of RNA molecules. The RNAInvBench paper (Cole et al., 2024) includes the pseudoknot-free and pseudoknot-inclusive benchmarks for RNA inverse design. It will be good to also compare against this benchmark.

**Questions:**

I have already left suggestions for further experiments that may help address some of the concerns in the weaknesses section. To summarize:
1. Does the model incorporate RNA modifications, cellular localization, or folding kinetics? Can the authors add some discussion on how might this enhance performance given the limited experimental training data or if they can provide some analysis for the same, if possible?
2. Apart from sequence recovery and perplexity, did the authors analyze structural stability and functional motifs that affect RNA viability?
3. Did the authors experiment with different RNA backbone representations, and what impact did this choice have on the results?
4. Can the authors analyze failure cases to improve understanding of gRNAde's limitations?
5. Have the authors evaluated how varying RNA sequence lengths affect performance?
6. Will the authors benchmark against the RNAInvBench paper (Cole et al., 2024) to assess the role of pseudoknots in their predictions as its importance has been highlighted as a crictical aspect to compare RNA inverse design models in RNAInvBench paper?

---

> ### Author Response · Authors · 2024-11-15
> **Thank you for your positive review and helpful suggestions**
>
> > 1. Does the model incorporate RNA modifications, cellular localization, or folding kinetics?
>
> No, gRNAde or any other inverse folding models that we are aware of (whether for proteins or RNA) do not consider modified nucleotides/residues nor cellular localization yet. There is very little data in the PDB with modified nucleotides/residues. Cellular localisation may not always be relevant for inverse folding depending on the application (eg. biotechnology/sensing applications). There is almost no data on folding kinetics of RNAs except studies of individual RNAs by J. Lucks' lab.
>
> In summary, we don’t have sufficient data for all 3 of those to be able to train ML models, as far as we are aware.
>
> > 2. On metrics apart from sequence recovery and perplexity
>
> We have included 2D and 3D structural self-consistency metrics in addition to the above. We have also discussed pros/cons of the various metrics in Section 2.3 and practical considerations of ranking designs in Appendix G.
>
> Regarding functional motifs – in practical RNA design applications, if specific functional/structural motifs are required as part of the design, gRNAde allows users to fix the identities of specific positions that participate in these motifs. The notebook [design.ipynb](https://anonymous.4open.science/r/geometric-rna-design/notebooks/design.ipynb) in the anonymous codebase shows this in cell #4.
>
> > 3. Different RNA backbone representations
>
> We did not try the consensus all-angle conformers nor modular string nomenclature.
>
> When we started development of gRNAde, we did preliminary experiments on the choice of backbone atoms to be used during featurization (Figure 13). Anecdotally, we found that using a 7-bead representation (over a 3-bead representation used in our final model) tended to fit training data better but produced the same or worse performance on the held-out test set. There is evidence from RNA structural biology literature that the 3-bead representation describes RNA backbones completely in most cases while reducing the size of the torsional space to prevent overfitting (Wadley et al., 2007).
>
> We haven’t formally ablated this design choice but thought you may find this interesting.
>
> > 4. Failure cases and limitations
>
> In experimental validation, we see that both gRNAde and Rosetta do not succeed on the GMP Riboswitch in its bound state (with its partner ligand) extracted from a longer mRNA, as well as the RNA component of the Telomerase enzyme (a protein-RNA complex). In both cases, the OpenKnot score of the Wildtype sequence also falls below 80. This suggests two failure modes:
> 1. **Lack of biological context**: Designing sequences for RNAs that have been extracted/removed from their broader biological context (a partner ligand and longer mRNA, or a protein-RNA complex) is more challenging than RNA-only design scenarios. We have noted this point as a limitation and avenue for future work in the Conclusion section, too.
> 2. **Dynamics**: The fact that the Wildtype sequence has tendency to not fold into the given target backbone structure suggests that the RNA may be more likely to be conformationally flexible (particularly known for Riboswitches). This was also a general observation from our computational benchmarks in Appendix C – the multi-state test set was significantly more challenging than the single-state test set in terms of all metrics.
>
> In summary, gRNAde and other structure-based design tools are best suited for tasks where folding stability is highly correlated with function. In addition to folding stability, natural RNAs are subject to other functional constraints, especially binding of partners, which we hope to incorporate into the next version of gRNAde.
>
> > 5. RNA sequence lengths affect on performance
>
> We have seen overall minor variations in sequence recovery with length. For instance, on the single-state benchmark:
> | Length range | gRNAde mean recovery | Rosetta mean recovery | RDesign mean recovery |
> |:---:|:---:|:---:|:---:|
> |(17.8, 46.2]|0.57 | 0.44 | 0.42|
> |(46.2, 74.4]| 0.51 | 0.45 | 0.42|
> |(74.4, 102.6]|0.54|0.44 | 0.45|
> |(102.6, 130.8]|0.44|0.44|0.39|
> |(130.8, 159.0]|0.61|0.53|0.56|
>
> > 6. Will the authors benchmark against the RNAInvBench?
>
> We weren’t aware of this work as it is pretty recent, but we have now cited the benchmark in the Related Work section in the revision. RNAInvBench focuses on 2D structure-based design (emphasis on base pairing) while our work focuses on 3D geometry (emphasis on non-canonical interactions and tertiary motifs).
>
> Any 3D RNA structure is folded into a specific shape due to tertiary interactions such as pseudoknots, kissing loops, or G-quadruplexes. Therefore:
> 1. All the RNA structures that we train and evaluate on from RNAsolo/PDB are already in the ‘pseudoknot-inclusive’ regime.
> 2. As far as we know, there are no ‘pseudoknot-free’ 3D RNA structures that we can use, b/c pseudoknots and other tertiary interactions are the drivers of RNA folding into 3D structures.

---

> > ### Comment · Reviewer_47od · 2024-11-21
> > **Thank you for your response**
> >
> > I thank the authors for their thorough responses. My concerns are addressed and I am happy to raise my score. I believe this paper makes a good contribution to the field,

---

> > > ### Author Response · Authors · 2024-11-21
> > >
> > > Thank you for the encouragement!

---

### Official Review · Reviewer_qxae · 2024-11-01

**Soundness:** 4
**Presentation:** 4
**Contribution:** 4
**Rating:** 8
**Confidence:** 2

**Summary:**

This paper develops an inverse folding model for RNA (structure -> sequence), analogous to proteinMPNN for protein sequences. Unlike prior work, it is able to consume multiple conformations as input. The model performs well compared to popular baselines such as Rosetta on a wide variety of evaluation tasks.

**Strengths:**

The paper is well written, well motivated, and the results are convincing. As a reviewer, I found the "COMPARISON TO CONTEMPORANEOUS WORK" section in the appendix highly useful. Thank you for including this. The paper has a remarkable number of results. The analysis in the appendix about model ablations would typically appear in the main paper, but there simply wasn't space.

**Weaknesses:**

I don't have enough background on RNA to understand some details of the evaluation, but I am familiar with analogous evaluations for proteins and am under the impression that the evaluation is quite comprehensive. It would be great if other reviewers with more RNA background could assess these details.

**Questions:**

I don't know much about the landscape of tools for this sort of design. My impression is that current versions of Rosetta do not support RNA design. Why is that? Is it because it was found that Rosetta did not perform well for it or simply that there wasn't support to maintain these capabilities in the software? Is it a strong enough baseline to be compare against?

It would be good to discuss the methodological similarities/differences vs. RDesign in the main paper too. Given the details in the appendix, I know this is a complex topic. Can you provide a short piece of text that you'd include in a new version of the main paper describing the overlap?

---

> ### Author Response · Authors · 2024-11-15
> **Thank you for your positive review**
>
> > My impression is that current versions of Rosetta do not support RNA design. Why is that? ... Is it a strong enough baseline to be compare against?
>
> As far as we understand, there has not been enough support to maintain the Rosetta RNA tools since early 2010s so they have not been included in the latest Rosetta builds. Recent deep learning-based tools like gRNAde represent a revival of interest in this problem: they are GPU-based/fast, more broadly accessible, and the designed RNAs have a higher experimental success rate than Rosetta! (We are now actively working on new biotechnology applications with experimental collaborators.)
>
> 3D structure-based RNA design has been a niche topic until recently, so we are not aware of other, better computational tools for this task. Thus, Rosetta is the state-of-the-art in physics-based RNA inverse folding to the best of our knowledge.
>
> Rosetta is also very well known in the broader biomolecule design community, which helps readers better contextualize our work.
>
>
> > ... methodological similarities/differences vs. RDesign in the main paper ... Can you provide a short piece of text that you'd include in a new version of the main paper describing the overlap?
>
> See page 7, line 330 onwards in the revised manuscript. We have included the following short piece of text summarizing key methodological differences: “RDesign is a concurrent deep learning-based 3D inverse folding model which uses invariant GNN layers and non-autoregressive decoding without sampling; see Appendix B for details.”

---

> > ### Author Response · Authors · 2024-11-23
> >
> > Does reading the other reviews and our responses improve your confidence in your assessment?

---

> ### Comment · Area_Chair_Qi9q · 2024-11-26
> **[ACTION NEEDED] Respond to author rebuttal**
>
> Dear Reviewer,
>
> We are reaching the end of the discussion period (November 26 at 11:59pm AoE): please check the author rebuttal and post your response at your earliest convenience. Please also update your review accordingly, even if your view of the paper has not changed (to acknowledge that you have read the rebuttal).
>
> Thank you, --Your AC

---

> > ### Author Response · Authors · 2024-12-01
> >
> > As the discussion period ends soon, let us know if our responses to yourself and the other reviewers have helped shape your assessment.

---

### Official Review · Reviewer_peU4 · 2024-11-02

**Soundness:** 4
**Presentation:** 4
**Contribution:** 4
**Rating:** 8
**Confidence:** 3

**Summary:**

The manuscript presents gRNAde, an approach for RNA inverse design conditioned on the 3D structure of RNA (conformational ensemble). The method uses geometric deep learning using a graph neural network which receives a set of backbone graphs derived from the 3D structure and predicts a set of RNA sequences (2D sequences of nucleotides (G,A,C,U).

The approach is similar to ProteinMPNN (Daupara 2022) that solved a similar inverse problem for proteins.

The two key related works are Rosetta (Leman 2020) and RDesign (Tan et al. 2023), a comparison to the latter is presented in pages 16-17. In particular, RDesign is another deep learning based method for inverse RNA design. Among various issues that the authors highlight in the appendix, it appears that the proposed method is open source, outperforms this baseline, and can offer sampling, which is attractive for producing multiple solutions.

**Strengths:**

The methodology is evaluated using both in silico and wet lab approaches. For in silico evaluation, three metrics (sequence recovery, self consistency, and perplexity) are reported. These metrics are described on page 16. The “wet lab” experiments are run on Eterna, an independent online computational RNA design platform, and showed an impressive performance improvement over Rosetta in the OpenKnot score.

The authors also create their own dataset using RNASolo (Adamczyk 2022). The authors should clarify this as an additional contribution if it is, or otherwise explain what are the modifications introduced to RNASolo. Will this new dataset also be released?

The paper is well written, polished and includes beautiful diagrams. The code is released open source (and added to the submission).

**Weaknesses:**

The authors should present a comprehensive technical comparison to other methods (e.g., Rosetta) not just RDesign (Tan et al. 2023) to enable better understanding of the proposed functionality. Specifically, please briefly summarize the technical components of Rosetta, and explain how gRNAde graph deep learning framework is different.

The authors should focus their discussion on the technical similarities and differences to previous work, without negativity (“This means that any design produced by RDesign is deterministic and not diverse, making it useless in practical scenarios. All recovery metric results reported in the RDesign paper are also incorrect due to this issue (at least based on how recovery is computed in protein design).“ (Page 16). There is no wet lab comparison to RDesign, and the authors should explain why this is not possible (e.g., using the re-engineered checkpoint that was used to compute some performance comparisons using recovery reported in Table 2 and Figure 2a).

**Questions:**

- Please explain what happens to the other non-successful examples from Eterna which are not successful. What is the intuition of why the proposed method (as well as Rosetta) does not work? Is this due to challenging structural motifs, sequence patterns, or other type of structural features? Are there inputs that gRNAde will be more suited for in comparison to other methods, and vice versa?

- Would the method generalize to unseen datasets/sources?

- In Figure 17, various properties of RNA sequences are presented for the dataset used (sequence length, number of structures/sequences, average pairwise RMSD per sequence, bivariate distribution for sequence length vs. avg. RMSD). Do these properties affect performance results when the model is evaluated?

- Are there any biases of the model, e.g., to certain pairs of nucleotides, e.g., CG pairs, see (Frnakenstein, Lyngsoe 2012)?

---

> ### Author Response · Authors · 2024-11-15
> **Thank you for your positive review and constructive comments**
>
> > ...briefly summarize the technical components of Rosetta
>
> We have updated the Related Work section’s description of Rosetta and how gRNAde is different in page 15, line 763 onwards.
>
> > ...focus their discussion on the technical similarities and differences to previous work, without negativity
>
> We have revised the statements in page 16, line 836 onwards to not be negative.
>
> > There is no wet lab comparison to RDesign, and the authors should explain why this is not possible
>
> The Eterna platform is independent to us and we are restricted in the number of designs we can submit and when they can be submitted. We have chosen to prioritize our model and have compared it to Rosetta (with the help of Eterna organizers). Rosetta outperforms RDesign in computational benchmarks, which further justifies prioritizing Rosetta.
>
> > Please explain what happens to the other non-successful examples from Eterna...
>
> Both gRNAde and Rosetta do not succeed on the GMP Riboswitch in bound state extracted from a longer mRNA, as well as the RNA component of the Telomerase enzyme (a protein-RNA complex). In both cases, the OpenKnot score of the Wildtype sequence also falls below 80. This suggests two failure modes:
> 1. **Lack of biological context**: Designing sequences for RNAs that have been extracted/removed from their broader biological context (a partner ligand and longer mRNA, or a protein-RNA complex) is more challenging than RNA-only design scenarios. We have noted this point as a limitation and avenue for future work in the Conclusion section, too.
> 2. **Dynamics**: The fact that the Wildtype sequence has tendency to not fold into the given target backbone structure suggests that the RNA may be more likely to be conformationally flexible (particularly known for Riboswitches). This was also a general observation from our computational benchmarks in Appendix C – the multi-state test set was significantly more challenging than the single-state test set in terms of all metrics.
>
> In summary, gRNAde and other structure-based design tools are best suited for tasks where folding stability is highly correlated with function. In addition to folding stability, natural RNAs are subject to other functional constraints, especially binding of partners, which we hope to incorporate into the next version of gRNAde.
>
> > Would the method generalize to unseen datasets/sources?
>
> Yes, we have been careful to evaluate gRNAde on test sets which evaluate for generalization to new RNA structures and folds unseen during training. Details are provided in Section 3, ‘Splits to evaluate generalization’. The Ribozyme in Section 4.3 is also unseen and gRNAde’s perplexity is useful for ranking its mutants’ fitness.
>
> > Do these properties affect performance results when the model is evaluated?
>
> Yes, the length and structural flexibility of an RNA backbone do impact gRNAde’s performance.
> - Flexibility: See Figure 3 (b) and the multi-state design benchmark – we find that performance is decreased in structurally flexible regions of RNA backbones, as characterized by increasing RMSD between the available states as well as by undergoing base pairing changes.
> - Length: We have seen overall minor variations in sequence recovery with length. For instance, on the single-state benchmark:
>
> | Length range | gRNAde mean recovery | Rosetta mean recovery | RDesign mean recovery |
> |:---:|:---:|:---:|:---:|
> |(17.8, 46.2]|0.57 | 0.44 | 0.42|
> |(46.2, 74.4]| 0.51 | 0.45 | 0.42|
> |(74.4, 102.6]|0.54|0.44 | 0.45|
> |(102.6, 130.8]|0.44|0.44|0.39|
> |(130.8, 159.0]|0.61|0.53|0.56|
>
>
> > Are there any biases of the model...
>
> The model reflects the biases present in its training data (from the PDB), which is ‘a feature and a bug’. Thus, designs from gRNAde are likely to be **‘PDB-like’ sequences** which are biased towards thermal stability. In practical scenarios, GC content can be enforced via logit biasing. This involves adding or removing a small bias term to the output logits during autoregressive decoding (the model samples the choice of base from its logits at each position during inference).
>
> The notebook [design.ipynb](https://anonymous.4open.science/r/geometric-rna-design/notebooks/design.ipynb) in the anonymous codebase actually shows how one can bias gRNAde’s sampling in practice – see cell #4.
>
>
> > Will this new dataset also be released?
>
> Yes, all the datasets are available in processed ML-ready format for anyone to develop new ideas on top of (RNAsolo provides raw PDB files). We have also released scripts which ingest the raw PDB files and prepare processed data. See ‘Downloading and Preparing Data’ section of [README.md](https://anonymous.4open.science/r/geometric-rna-design/README.md) in the anonymized code. Our datasets have already been used by other researchers. We initially didn’t consider this major enough to be listed as a contribution, but thanks for noticing this.

---

> > ### Comment · Reviewer_peU4 · 2024-11-19
> > **Thank you**
> >
> > Thank you for clarifying the technical comparison to previous work and effect of various parameters. I have updated my score.

---

> > > ### Author Response · Authors · 2024-11-19
> > >
> > > Thank you for updating your score!

---

### Meta-Review · Area_Chair_Qi9q · 2024-12-20

**Metareview:**

This paper introduces gRNAde, a method for RNA sequence inverse design given 3D RNA structure information.

All reviewers agree that this is a solid paper with strong results. Similar techniques exist for general protein design (e.g. ProteinMPNN), but the paper adds sufficient value to the community and is generally of high quality, warranting acceptance.

**Additional Comments On Reviewer Discussion:**

Reviewer consensus did not change during discussion.

---

### Decision · Program_Chairs · 2025-01-22

Accept (Spotlight)